# Small emission sources in aggregate disproportionately account for a large majority of total methane emissions from the US oil and gas sector

James P. Williams[1], Mark Omara[1,2], Anthony Himmelberger[2], Daniel Zavala-Araiza[1], Katlyn MacKay[1], Joshua Benmergui[1,2,3], Maryann Sargent[3], Steven C. Wofsy[3], Steven P. Hamburg[1,2], Ritesh Gautam[1,2]

[1]Environmental Defense Fund, New York, NY, USA 10010
[2]MethaneSAT, LLC, Austin, TX, USA 78701
[3]Harvard University, Cambridge, MA, USA 02138

*Correspondence to:* James P. Williams (jamwilliams@edf.org), Ritesh Gautam (rgautam@edf.org)

**Abstract.** Reducing methane emissions from the oil and gas (oil/gas) sector has been identified as a critically important global strategy for reducing near-term climate warming. Recent measurements, especially by satellite and aerial remote sensing, underscore the importance of targeting the small number of facilities emitting methane at high rates (i.e., "super-emitters") for measurement and mitigation. However, the contributions from individual oil/gas facilities emitting at low emission rates that are often undetected are poorly understood, especially in the context of total national- and regional-level estimates. In this work, we compile empirical measurements gathered using methods with low limits of detection to develop facility-level estimates of total methane emissions from the continental United States (CONUS) midstream and upstream oil/gas sector for 2021. We find that 70% (95% confidence intervals: 61-81%) of the total 14.6 (12.7-16.8) Tg/yr oil/gas methane emissions in the CONUS for the year 2021 originate from facilities emitting <100 kg/hr, and 30% (26-34%) and ~80% (68-90%) from facilities emitting <10 kg/hr and <200 kg/hr, respectively. While there is variability among the emission distribution curves for different oil/gas production basins, facilities with low emissions are consistently found to account for the majority of total basin emissions (i.e., range of 60% - 86% of total basin emissions from facilities emitting <100 kg/hr). We estimate that production well sites were responsible for 70% of regional oil/gas methane emissions, from which we find the well sites that accounted for only 10% of national oil and gas production in 2021, disproportionately accounted for 67-90% of the total well site emissions. Our results are also in broad agreement with data obtained from several independent aerial remote sensing campaigns (e.g., MethaneAIR, Bridger Gas Mapping LiDAR, AVIRIS-NG, and Global Airborne Observatory) across 5-8 major oil/gas basins. Our findings highlight the importance of accounting for the significant contribution of small emission sources to total oil/gas methane emissions. While reducing emissions from high-emitting facilities is important, it is not sufficient for the overall mitigation of methane emissions from the oil and gas sector which according to this study is dominated by small emission sources across the US. Tracking changes in emissions over time and designing effective mitigation policies should consider the large contribution of small methane sources to total emissions.

38

## 1 Introduction

Methane is a short-lived but powerful greenhouse gas with a global warming potential more than 80 times stronger than carbon dioxide ($CO_2$) over 20 years (AR6 Synthesis Report: Climate Change 2023, 2024). Therefore, the reduction of methane emissions has become a key goal to achieve rapid climate mitigation in the short term (Ocko et al., 2021). In North America, one of the largest sources of methane emissions originates from the oil and gas (oil/gas) sector, with most emissions originating from the production (i.e., upstream) and transportation/storage (i.e., midstream) sectors (Alvarez et al., 2018). Multiple studies, especially over the past decade, have focused on the quantification of methane sources from the oil/gas sector, with particular emphasis on the continental United States (CONUS) (Alvarez et al., 2018; de Gouw et al., 2020; Omara et al., 2018; Lu et al., 2022; Zhang et al., 2020; Shen et al., 2022; Cusworth et al., 2022; Nesser et al., 2023; Brandt et al., 2016; Duren et al., 2019; Maasakkers et al., 2021; Lu et al., 2023; Worden et al., 2022). Several studies have recognized the importance of a small percentage of high-emitting sites (i.e. "super-emitters") and reported them as accounting for a large fraction of total methane emissions (Brandt et al., 2016; Cusworth et al., 2022; Duren et al., 2019; Sherwin et al., 2024). The emission rate thresholds that characterize these super-emitting facilities are critical information for methane measurement platforms, especially remote sensing technologies focused on detecting high-emitting point sources. Aerial and satellite remote sensing technologies have enabled more frequent monitoring of emissions from oil and gas sites and rapid mapping of large areas, although they face limitations in detection sensitivity. Despite the improved ability to locate and quantify emissions from high-emitting sites, there has been considerable lack of understanding about the characteristics of low methane emitting facilities, especially those emitting at rates below the limits of detection (LOD) of most point-source detection remote sensing platforms, and their contributions to total oil/gas methane emissions.

While some studies offer important yet limited insights into the contributions of different lower-emitting infrastructure from the CONUS oil/gas sector, there is a lack of understanding about their overall contribution to the total sectoral regional and national scale emissions. A recent study by Xia et al. (2024) combined aerial remote sensing data from Bridger Gas Mapping LiDAR (Bridger GML) in four oil/gas basins supplemented with component-level modeling for facilities emitting below the Bridger GML LOD and found significantly more emission sources in the 1 – 10 kg/hr range when compared to the emission distribution used by the EPA (Xia et al., 2024). In a study focused on production well sites in the CONUS, the main source of methane emissions from the oil/gas sector (Alvarez et al., 2018; Omara et al., 2018; Rutherford et al., 2021), Omara et al. (2018) found that 90% of total methane emissions from producing well sites came from those emitting at rates <100 kg/hr. A follow-up study by Omara et al. (2022) highlights that the total methane emissions from low-producing well sites producing less than 15 boe/day (i.e., 1 Mcf = 1,000 cubic feet of natural gas = 19.2 kg of methane at 15.6 ºC and 1 atmosphere; 1 boe = 1 barrel of oil equivalent = 6 Mcf; assumed methane content in natural gas of 80%), which comprise 80% of

all producing well sites in the CONUS, were responsible for nearly half of all methane emissions from the oil/gas production sector. Kunkel et al. (2023) observed that the use of the Bridger GML remote sensing platform with an LOD of 3 kg/hr, combined with prior Carbon Mapper detections in a section of the Permian basin showed a significant contribution from sources below the listed LOD of Carbon Mapper of 10 kg/hr. Cusworth et al. (2022) found that 35% of total methane emissions (including non-oil/gas sources) from several major oil/gas producing basins (other than the Appalachian basin) in the CONUS come from facilities emitting >10 kg/hr, indicating that 65% of emissions come from facilities emitting <10 kg/hr. Although these studies using independent measurement platforms provide new emerging insights about the importance of low methane emitting oil/gas facilities, there generally remains a lack of quantitative assessment of the relative fractions of emissions originating from different emission rate thresholds aggregated over individual oil/gas basins as well as at a national scale.

There are a variety of different methane quantification methods that differ in terms of their spatial resolution of sources, logistical constraints, costs of implementation, and their LODs. Measurement method sensitivities and LODs have important policy implications. For example, the Environmental Protection Agency (EPA) recently finalized regulations that define a "super-emitter event" as an emission rate threshold of 100 kg/hr or greater (Standards of Performance for New, Reconstructed, and Modified Sources and Emissions Guidelines for Existing Sources: Oil and Natural Gas Sector Climate Review, 2024), albeit without clear information on what percentage of total regional emissions are captured within this definition. Satellite and aerial remote sensing methods have point source LODs that range anywhere from 1-3 kg/hr for Bridger's airborne GML (Johnson et al., 2021; Kunkel et al., 2023; Thorpe et al., 2024; Xia et al., 2024) to ~200 kg/hr for GHGSat (Sherwin et al., 2023). In contrast, ground-based measurement methods such as OTM-33a and tracer release have LODs <1 kg/hr (Fox et al., 2019). A study by Ravikumar et al. (2018) using the Fugitive Emissions Abatement Simulation Toolkit (FEAST) suggests that a method with a LOD of 0.1-1 kg/hr would sufficiently capture all emissions from the oil/gas sector, whereas the ability to quantify emissions below this threshold would not lead to any significant increases in mitigation. Ultimately, there is a need for clarification in the total percentage contribution of emissions originating from a given emission rate threshold, which requires characterizing entire emissions distributions, not only the high emitters.

In this work, we create and analyze measurement-based methane emission rate distributions of US upstream and midstream oil/gas facilities to determine the percentage contributions of different emission rate thresholds to total methane emissions. First, we use empirical measurements gathered from ground-based sampling platforms to develop a bottom-up facility-based model to estimate methane emissions for upstream and midstream facilities in the continental US (CONUS) for 2021. Next, we aggregate our facility-level, population-based data to determine the national- and basin-level contributions of methane emissions originating from facilities emitting at different emission rate thresholds, in addition to comparisons to aerial-remote sensing platforms. Finally, we break down the emission distribution curves by facility category to analyze how the percentage contributions of total emissions vary across facility types.

## 2 Materials and methods

### 2.1 Empirical measurements

We compile 1,901 facility-level methane emission rate measurements from 16 studies (Brantley et al., 2014; Caulton et al., 2019; Deighton et al., 2020; Goetz et al., 2015; Lan et al., 2015; Mitchell et al., 2015; Omara et al., 2016, 2018; Rella et al., 2015; Riddick et al., 2019; Robertson et al., 2017, 2020; Subramanian et al., 2015; Yacovitch et al., 2015; Zhou et al., 2021; Zimmerle et al., 2020) that use ground-based site/facility level and source/component level measurement methods with low LOD's of ~0.1 kg/hr. Most (i.e., 85%) of empirical measurements we use in this work were gathered using ground-based mobile laboratories that quantified methane emissions at the site/facility level using either tracer-based releases, the EPA Other Test Method (OTM-33a), or Gaussian plume transport modeling (Fox et al., 2019) (Table S2). The remaining 15% of empirical measurements we use (Deighton et al., 2020; Riddick et al., 2019; Zimmerle et al., 2020) are ground-based methods that aggregated source/component-level HiFlow sampling or static/dynamic chamber measurements, which could mean that other on-site emission sources were not quantified during measurement and overall emission rate estimates are conservative. Only one study was excluded from our analysis (ERG, 2011) due to a combination of age and a focus on component-level measurements.

The compiled empirical measurements target a variety of production well sites and/or midstream facilities across at least nine oil/gas-producing basins in the CONUS (Table S3). For all facility categories (i.e., production well sites, gathering and boosting compressor stations, transmission and storage compressor stations, and processing plants), we prioritize datasets of randomly sampled sites that include measurements below the method's LOD or reported as zero emissions, except for measurements from two studies (Brantley et al., 2014; Lan et al., 2015) which we discuss later in Section 2.3. Additionally, for production well site measurements, we focus only on data that provide facility-level gas production data for the date/month of measurement. Our compiled dataset of measurements includes both routine intentional (e.g., venting from pneumatic devices) and non-intentional (e.g., malfunctioning equipment and/or leaks from valves, connectors, flanges, etc) emissions, and while we remove any measurements attributed to high emitting intermittent events such as flowbacks and liquids unloadings if that information is present, we cannot fully discount that emissions from these high-emitting intermittent sources are included in our compiled dataset. Furthermore, we remove any empirical measurement data associated with flaring emissions, which are treated separately as discussed below, if that information is provided in the empirical data.

We categorize the empirical measurements by facility category as production well sites, gathering and boosting compressor stations (G&B compressors), transmission and storage compressor stations (T&S compressors), or processing plants. We group the empirical measurements from production well sites into six production bins based on gross average daily gas production as reported in individual studies. We use gross daily average gas production data instead of oil and gas production data for two reasons: 1) the limited availability of facility-level oil production data provided from empirical measurement studies; and 2) the established relationship between gas production and

emission rates observed in previous work (Omara et al., 2018, 2022, 2024). The gas production ranges of the
production bins (Fig. 1) are chosen to evenly distribute empirical measurements above the method LOD to all six
production bins. This categorization creates nine distinct facility categories: G&B compressors, T&S compressors,
processing plants, and six groups of production well sites. We further classify the nine distinct facility categories
into five primary facility categories: low-production well sites which produce combined oil and gas <15 boe/day
(i.e., 0.13 kt of methane production per year), non-low-producing well sites which produce ≥15 boe/day, processing
plants, G&B compressors, and T&S compressors. In addition to these facility categorizations, we also include
Visible Infrared Imaging Radiometer Suite (VIIRS) flare detections and flared gas volume estimates in our analysis,
which are treated as an independent methane source since flares can be located on multiple facility categories across
the upstream and midstream oil/gas sectors.

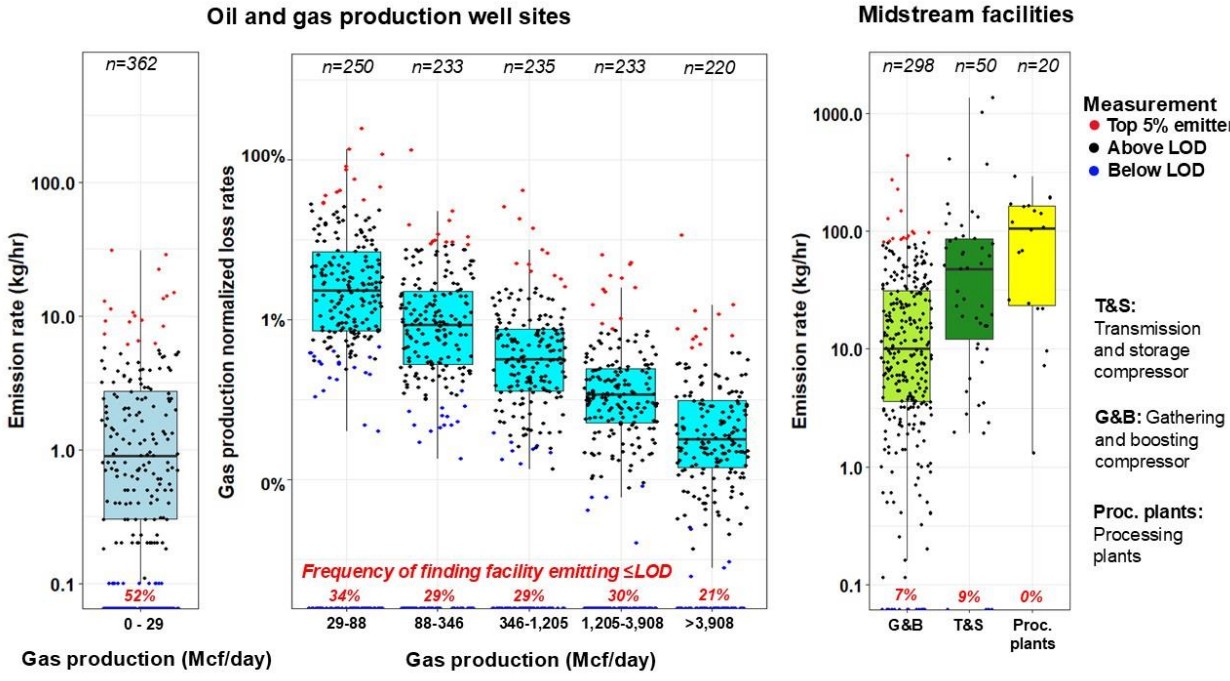


**Figure 1:** Facility-level empirical measurement data distributed by different distinct facility categories for production well sites
(left) and midstream facilities (right). Individual measurements are shown for each box plot and colored according to their
emission rate status for that facility category, where blue points are considered non-detectable emissions below an emission rate
threshold of ≤0.1 kg/hr/facility which is the method LOD we use, black points are measurements above our method LOD but
below the top 5% emitter category, and red points are the top 5% of empirical emission rates or loss rates for that facility
category. The number of empirical measurements available for each facility category is denoted at the top of each boxplot. The
estimated mean frequency of finding a facility emitting below the method LOD is shown in inset red text at the bottom of each
boxplot  We show absolute emission rates (kg/hr) rather than normalized loss rates (%) for the lowest cohort of production well
sites due to the reasoning presented in Section 2.3. Unit conversions: 1 Mcf = 1,000 cubic feet of natural gas = 19.2 kg of
methane at 15.6 °C and 1 atmosphere; 1 boe = 1 barrel of oil equivalent = 6 Mcf; assumed methane content in natural gas of 80%.

**2.2 Activity data**

We use activity data (i.e., number of facilities and spatial locations) for actively producing wells in 2021
provided by Enverus for the CONUS. We calculate both the annual averaged daily gross gas production, and oil and
gas production for each producing well using the number of producing days and total annual oil and gas production
data provided by Enverus. We convert production wells to production well sites by spatially aggregating individual
wells within 25-meter (vertical wells) or 50-meter (horizontal wells) distances from each other and separately
merging their combined oil and gas production and gas production, and converting these production values to a mass
equivalent production rate in kg/hr of methane (i.e., 1 Mcf = 1,000 cubic feet of natural gas = 19.2 kg of methane at
15.6 °C and 1 atmosphere; 1 boe = 1 barrel of oil equivalent = 6 Mcf; assumed methane content in natural gas of
80%), similar to previous approaches (Omara et al., 2018).
We acquire activity data for operational transmission and storage (T&S) and gathering and boosting (G&B)
compressor stations and processing plants from Enverus for 2021 for the CONUS, which was further supplemented
by additional data from the Oil and Gas Infrastructure Mapping (OGIM) database published in Omara et al. (2023).
We filter data for these midstream facilities to include only active facilities in the year 2021. For VIIRS flare
detections, we use the 2021 natural gas flared volume estimates based on natural gas flaring detections provided by
the VIIRS instruments installed aboard satellite platforms which have a 750x750 meter source resolution (NOAA-20
and Suomi National Polar-orbiting Partnership) (Elvidge et al., 2015). In terms of potential double-counting between
the VIIRS flare detections and the empirical measurements we use in this work, the majority of VIIRS detections are
in the Permian, Bakken, and Eagle Ford oil/gas basins (i.e., 86% of total VIIRS detections) which corresponds to a
small number of our empirical measurement data (Table S3) (Plant et al., 2022). However, the limited availability of
spatial coordinates for our empirical measurements restricts our ability to perform a direct comparison to exclude
overlapping/proximal VIIRS detections and our facility-level empirical measurements. Therefore, we do
acknowledge that there is a possibility of double counting between our empirical measurement data and the VIIRS
flare detections, but we expect the degree of overlap to be low.
**2.3 Facility-level methane emission inventory**
We calculate annual methane emissions from all facility categories (i.e., six production bins of production well
sites, T&S compressor stations, G&B compressor stations, processing plants, and VIIRS flare detections) using a
multi-step probabilistic modeling approach adapted from multiple studies (Omara et al., 2018, 2022; Plant et al.,
2022) (Fig. 2). Briefly, for each individual facility and VIIRS flare detection in the CONUS for 2021, we estimate an
annually averaged methane emission rate using empirical measurement data, and consequently the cumulative
distribution of methane emission rates from the aggregation of these individual emission rates. Each emission rate
estimate is indexed according to the corresponding replicate (n=500), and we use these repetitions to determine
uncertainty for the cumulative methane emission distribution curves. The detailed steps of this process for all facility
categories and VIIRS flare detections are described below.
For the highest five gas production bins of producing well sites ranging from 29 to >3,908 Mcf/day (or 0.2 to
>27 kt of methane production per year, Figure 1), we use gross gas production normalized loss rates to model the
distributions used to calculate methane emission rates from Eq. (1), where the: *Loss rate* is the fraction of emitted
gas relative to gas production, the *emission rate* is rate of methane emitted from a facility in kilograms per hour,
$\sigma_{CH4}$ is the methane content of the emitted gas which we assume to be 80%, and the *gas production* is the mass
equivalent of natural gas produced in kilograms per hour at 1 atmosphere and 15.6 °C (1 Mcf = 1,000 cubic feet of
natural gas = 19.2 kg of methane at 15.6 °C and 1 atmosphere; 1 boe = 1 barrel of oil equivalent = 6 Mcf). For the
lowest well site gas production bin of 0 to 29 Mcf/day (i.e, 0 to 0.2 kt of methane production per year) and
midstream facilities, we use the empirical absolute methane emission rate data as is. This approach is partly based
on the methods used by Omara et al. (2022) for the low production well site category, which exploits a weak
relationship between gross gas production data (which is most accessible in empirical measurement studies) and
absolute emission rates to better extrapolate emissions to the entire population of production well sites in the
CONUS.
$$Loss\ rate = \frac{Emission\ rate\ \left[\frac{kg}{hr}\right]}{\sigma_{CH_4} \times Gas\ production\ \left[\frac{kg}{hr}\right]} \quad (1)$$
For our estimation of facility-level emission rates, we break down the modeling process into two separate steps:
the first determines whether a randomly selected facility is emitting methane above our method LOD of ≤0.1
kg/hr/facility, and the second determines the associated methane emission rate for that individual facility. To test the
sensitivity of our method to the selection of the method LOD, we also perform an additional sensitivity analysis for
other method LODs (Fig. S8). The processes outlined below are all specific to each of our nine facility categories.
Brantley et al. (2014) and Lan et al. (2015) are excluded from this first step since they do not include measurements
below the method LOD but include valuable data on well site emission rates with associated well site production
data. To determine whether a facility is emitting methane above the method LOD threshold in our estimates, we first
use bootstrapping with replacement (n=1,000) of our empirical measurement data to simulate the frequency of
finding an individual facility emitting methane above the method LOD (i.e., ≤0.1 kg/hr/facility), which we call an
"emitting facility" or "emitter" herein (Fig. 2). The results of the bootstrapping procedure represent a normal
probability distribution from which we estimate the frequency of finding an emitting facility (i.e., above the method
LOD) with associated uncertainty bounds. Next, we remove the empirical measurements below the LOD and use
bootstrapping with replacement (n=1,000) on the above LOD empirical measurements to determine the probability
of an emitting facility being in the top 5% (i.e., 95[th] percentile or above of empirical measurement data) or bottom
95% (i.e., 95[th] percentile or below the empirical measurement data) of emitters, except for processing plants and
T&S compressors which had too few measurements (n=20 and n=50 respectively) to distinguish between the top 5%
and bottom 95% of emission or loss rates. Similar to the process of determining the frequency of finding an emitting
facility, we use the results of the bootstrapping to develop a normal probability distribution that classifies an
emitting facility as either a top 5% or bottom 95% emitter. This pseudo-random selection of a top 5% emitter within
each facility category accounts for the functional definition of abnormally large emissions (i.e., super-emitters) that
can be observed in all facility categories (including well sites in different production bins) (Zavala-Araiza et al.
2015, Brandt et al. 2016). We fit the results of the bootstrapping to two normal distributions: one for the top 5% of
emitters and one for the bottom 95% of emitters. We use the associated parameters of each normal distribution to
randomly determine whether a facility is emitting in the top 5% or bottom 95% of emitters. These steps are repeated
for each facility for each facility category in the CONUS.
At the end of the first step of this facility-level modeling process, all facilities in the CONUS are classified as
either a: bottom 95% emitter, top 5% emitter, or below the method LOD. Loss rates are used to calculate emission
rates for the top five highest production bins of well sites, whereas we directly estimate methane emission rates for
the well sites in the lower production cohort (Fig. 1), and for midstream facilities excluding VIIRS flare detections.
For facilities classified as the top 5% and bottom 95% of emitters, we estimate their methane emissions by first
fitting a lognormal distribution to the empirical measurement data, including measurements from Brantley et al.
(2014) and Lan et al. (2015), of either the gas production normalized loss rates or methane emission rates (Eq. 1),
depending on the facility category. Next, we use the parameters of the modeled distributions to randomly assign
either an emission or loss rate to a randomly selected facility (n=500), depending on its emitter status and facility
category. We test each estimated methane emission distribution to the associated empirical measurements and find a
good fit for all facility categories (Table S6). To account for facilities emitting below the method LOD, we
randomly assign an emission rate from re-sampling our dataset of empirical measurements below the method LOD
for that facility category. Finally, once all facilities are assigned an emission rate, we compile the ensemble of
emission distributions to develop facility-level emission distribution curves and total regional oil/gas methane
emissions for the CONUS in 2021.
For all VIIRS flares detections, we use the total reported volumes of gas flared for 2021 from flares detected
using the VIIRS instrument (Elvidge et al. 2015) multiplied by the observed flare destruction efficiencies and
percentage of unlit flares from Plant et al. (2022) to calculate annual methane emission rates from this source. As
previously stated, our empirical measurements are largely located outside of oil/gas basins where the majority of
VIIRS flare detections are located (i.e. Permian, Eagle Ford, and Bakken), but we cannot discount the possibility
that there are instances of double-counting flares measured via our ground-based empirical data and those detected
by VIIRS. For each VIIRS flare detection, we randomly determine whether it is an unlit or lit flare based on the
basin-specific percentages of unlit flares reported by Plant et al. (2022). If a flare is determined to be lit, we use the
corresponding basin-specific observed destruction removal efficiencies as reported by Plant et al. (2022) multiplied
by the corresponding annual total volume of gas flared and convert to an emission rate. The basin-specific observed
destruction removal efficiencies are estimated through a fitted normal distribution using the mean and standard
deviations modeled from the 95% confidence intervals presented in Plant et al. (2022). If a flare is determined to be
unlit, we use a destruction removal efficiency of 0%. For VIIRS flare detections located outside of the Bakken,
Eagle Ford, and Permian basins, we used the total CONUS averaged destruction removal efficiencies of 95.2%
(95% confidence interval: 94.3 – 95.9%) and percentage of unlit flares of 4.1% as reported by Plant et al. (2022).

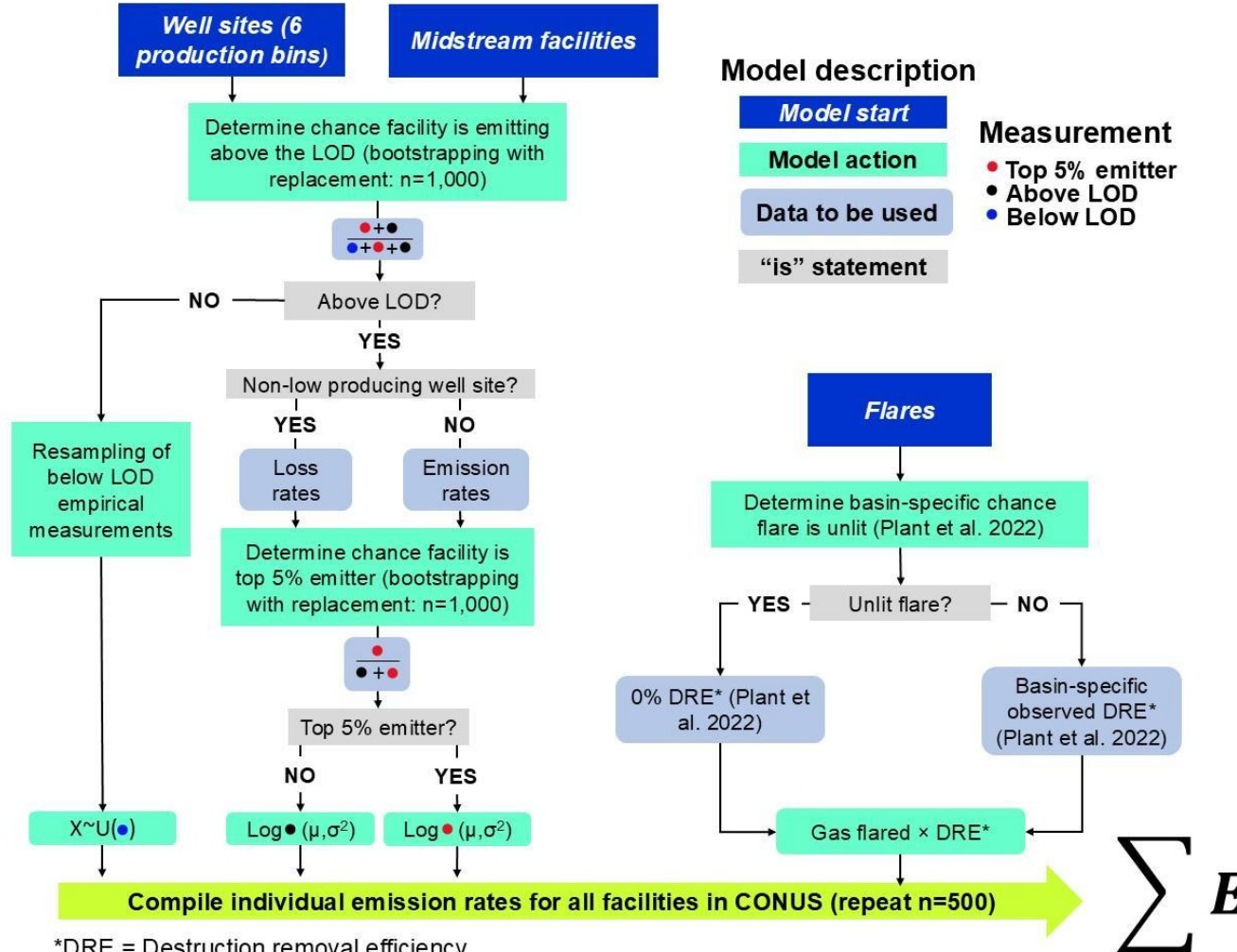


**Figure 2:** Flowchart describing the facility-level estimates, with steps colored according to the specific process and data being
used. We note that methane emission rates for flares are calculated using a separate approach from that of production well sites
and midstream facilities. Processing plants and T&S compressors are excluded from the determination of whether a facility is a
top 5% emitter due to a lack of available empirical measurement data.

**2.4 Extrapolation to smaller spatial boundaries**

We perform several comparisons of our estimated emission distribution curves and total aggregated
emissions to estimates from aerial and satellite remote-sensing studies. To perform these comparisons, we restrict
our estimates and the results from other aerial/satellite studies to spatial domains of interest (e.g., an oil/gas basin
boundary or the overflown domain from an aerial sampling campaign), and to specifically compare estimates of
oil/gas methane emissions from the facility categories we are investigating in this work. For comparisons to satellite
remote-sensing studies, we prioritize national-level satellite inversions that estimate methane emissions from the
CONUS that include spatially explicit maps of methane emission inversions specifically for oil/gas sources. We join
the spatially explicit satellite inversions of methane emissions to the top twelve producing oil/gas basin boundaries

in the CONUS, in addition to their national-level inversions which we also use for national comparisons. Since our
facility-level model includes geo-located activity data (i.e., facility coordinates), we can estimate facility-level
methane emissions distributions and estimate total methane emissions for any spatial boundary in the CONUS by
spatially joining facilities within a target boundary. Spatial variability in our facility-level estimates is driven by two
main factors: counts of facilities and facility types, and averaged annual production characteristics. Due to
constraints on data availability, we do not constrain our available empirical measurement data to the specific regions
where they were gathered (Table S3). We tested the sensitivity of excluding empirical measurements gathered from
specific oil/gas on the national emission distribution curves and total national methane emissions and found no
significant variation (Fig. S9). Due to a lack of data availability, we do not have sufficient spatial information from
empirical measurements of G&B compressors, T&S compressors, and processing plants to test for basin-level
differences in empirical measurement data.

For comparisons to aerial remote sensing studies/results, we prioritize studies that include both measured

point sources (i.e., oil/gas methane sources that are above the LOD of the aerial remote sensing measurement
platform), estimates of total regional oil/gas emissions, and descriptions/outlines of the surveyed spatial domains
which are required for these comparisons. Based on these criteria, we compare our estimated emissions to those
from  peer-reviewed studies (Cusworth et al., 2022; Kunkel et al., 2023; Xia et al., 2024) and the results of research
flights from MethaneAIR in the Permian and Uinta oil/gas basins (Omara et al., 2024; Chan Miller et al., 2023;
Chulakadabba et al., 2023; MethaneAIR, 2024), with discussion in later sections on a recent study by Sherwin et al.
(2024). In all cases, we estimate facility-level methane emissions within the spatial domains outlined by the aerial
remote sensing studies to estimate region-specific methane emission distribution curves, use the relevant method
limits of detection to characterize emission rate thresholds valid for comparison, and subtract any emission unrelated
to the facility types we characterize (Chen et al., 2024). In the case of Cusworth et al. 2022, we infer the spatial
domains by georeferencing figures from their studies using the georeferencer tool QGIS (v.3.34.2-Prizen). We
compare our spatially joined facility-level emission distributions to the percentage of emissions contributed from
facilities emitting below discrete methane emission rate thresholds for all four aerial remote sensing studies, and to
the continuous cumulative methane emissions distribution curves from Bridger GML surveys (Kunkel et al., 2023;
Xia et al., 2024).

Each aerial remote sensing campaign utilizes independent methods to estimate their percentage

contributions from small methane sources, which in some cases requires additional analysis of the aerial remote
sensing results. For our analysis of continuous methane emissions distribution curves from the Bridger GML
campaigns (Kunkel et al., 2023; Xia et al., 2024), we restrict our analysis to estimated emission rates >3 kg/hr,
which is the approximate LOD of the Bridger GML remote sensing platform. For MethaneAIR, we use the
percentage of area emissions (i.e., diffuse area methane sources) relative to the total methane emissions for the
spatial boundary, which roughly corresponds to all emissions <200 kg/hr (i.e. effectively those emissions below the
point source detection limit of MethaneAIR that flew in multiple campaigns in the US at 40,000ft above ground
level (Chulakadabba et al., 2023)). MethaneAIR characterizes the total regional emissions including the spatial area
emissions at high resolution using a geostatistical inverse modeling framework (Miller et al., 2013) while ingesting
high-emitting point source information in the inversion (Chulakadabba et al., 2023; Omara et al., 2024).  For
Cusworth et al. (2022), we analyze all campaigns by subtracting both aerially detected pipeline emissions and all
non-oil/gas emissions (e.g., wastewater, landfills, agriculture), since our study is focused solely on upstream and
midstream oil/gas sources. In addition, we subtract emissions from pipelines and non-oil/gas sources emitting below
aerial detection limits (i.e., TROPOMI inversions subtracted by aerially detected emissions) by estimating the
relative fractions of pipeline and non-oil/gas sources from the aerial detections, with the assumption that these
fractions are representative (Table S4). However, this process can introduce additional uncertainties in our
comparisons, especially for campaigns where 50% or more of aerially detected emissions were from pipelines or
non-oil/gas sources.

We account for the intermittency of detected methane sources with <3 overpasses in Cusworth et al.

(2022) by resampling with replacement (n=1,000) the source persistence of methane sources with ≥3 overpasses for
the same campaign, which is consistent with their methodology. We calculate the percentage contributions of low
emitting sources in Cusworth et al. (2022) using Eq. (2): where $\%E_{[<x]}$ is the percentage of total oil/gas methane
emissions below an emission rate threshold $x$ (kg/hr), $T$ is the total area emissions measured via TROPOMI
inversions (kg/hr), and $P_{[>x]}$ is the sum of point source emissions above the emission rate threshold $x$ (kg/hr).
$$\%E_{[<x]} = 1 - \frac{P_{[>x]}}{T} \qquad\qquad (2)$$
**2.5 Uncertainty calculations**

Our emission distributions based on facility-level estimates incorporate uncertainty through several steps, such

as the: probabilistic distributions of a select facility being a top 5%, bottom 95% emitter, or facility emitting below
the LOD; emission rate and loss rate distributions produced from facility-level empirical measurements; and flaring
combustion efficiencies. In addition, we incorporate uncertainties from the empirical measurements into our facility-
level model by simulating new empirical emission rates based on the associated method uncertainties. At the
beginning of each of the 500 model iterations, we use the reported empirical methane emission rate data and
estimate a new emission rate using a normal distribution with the mean as the initial reported emission rate and the
standard deviation as a percentage of the mean value. These measurement uncertainties (i.e., 1-sigma) are chosen
based on the measurement methodology using the lower percentage uncertainty ranges provided by Fox et al. (2019)
for facilities measured via the OTM-33a (±25%), Gaussian plume dispersion (±50%), and tracer release (±20%)
methods. For HiFlow sampler measurements, we use an uncertainty range of ±16% (Riddick et al., 2022), and for
chamber-based measurements, we use ±14% (Williams et al., 2023). Therefore, each model iteration incorporates a
unique suite of empirical measurement data based on the initially reported emissions and their associated
uncertainties, which in turn impacts the probabilistic modeling of the chance of a facility emitting below the method
LOD, the empirical data is used to determine the parameters of the lognormal distributions of loss rates and emission
rates, and the ranges of the production bins. To calculate the cumulative uncertainty of our facility-level model
estimates, we estimate 500 methane emission distributions and aggregate the $2.5^{th}$ and $97.5^{th}$ percentiles of our seven
primary facility categories (i.e., low and non-low producing well sites, G&B compressors, T&S compressors, and
processing plants), which include lit and unlit VIIRS flare detection emissions, to determine our 95% confidence
intervals. This process is repeated for all simulations at the national-, basin-, and aerial remote sensing boundary
levels. For uncertainty calculations in satellite- and aerial-remote sensing studies we use for comparisons, we present
the reported 95% confidence intervals, if available.

## 3 Results


## 3.1 Distribution of emission rates at the national scale



Based on the results from our facility-level model estimates, we estimate that 70% (95% confidence

interval: 61-81%) of total methane emissions from the upstream/midstream sector in the CONUS for 2021 originate
from facilities emitting methane at rates <100 kg/hr (Fig. 3). For other emission rate thresholds, we find that 30%
(26-34%) of total emissions come from facilities emitting <10 kg/hr, which corresponds to the lower thresholds of
aircraft-based aerial remote sensing studies (Cusworth et al., 2022; Johnson et al., 2021; Kunkel et al., 2023; Thorpe
et al., 2024; Xia et al., 2024), and 79% (68-90%) of total emissions come from facilities emitting <200 kg/hr. We
find that the emission rate threshold corresponding to 50% of cumulative methane emissions from
upstream/midstream facilities in the CONUS for year 2021 is 25 kg/hr (19-33 kg/hr). These results suggest that a
large majority of oil/gas emissions in the CONUS are not detectable by existing satellite remote-sensing point
source imagers (Sherwin et al., 2023).

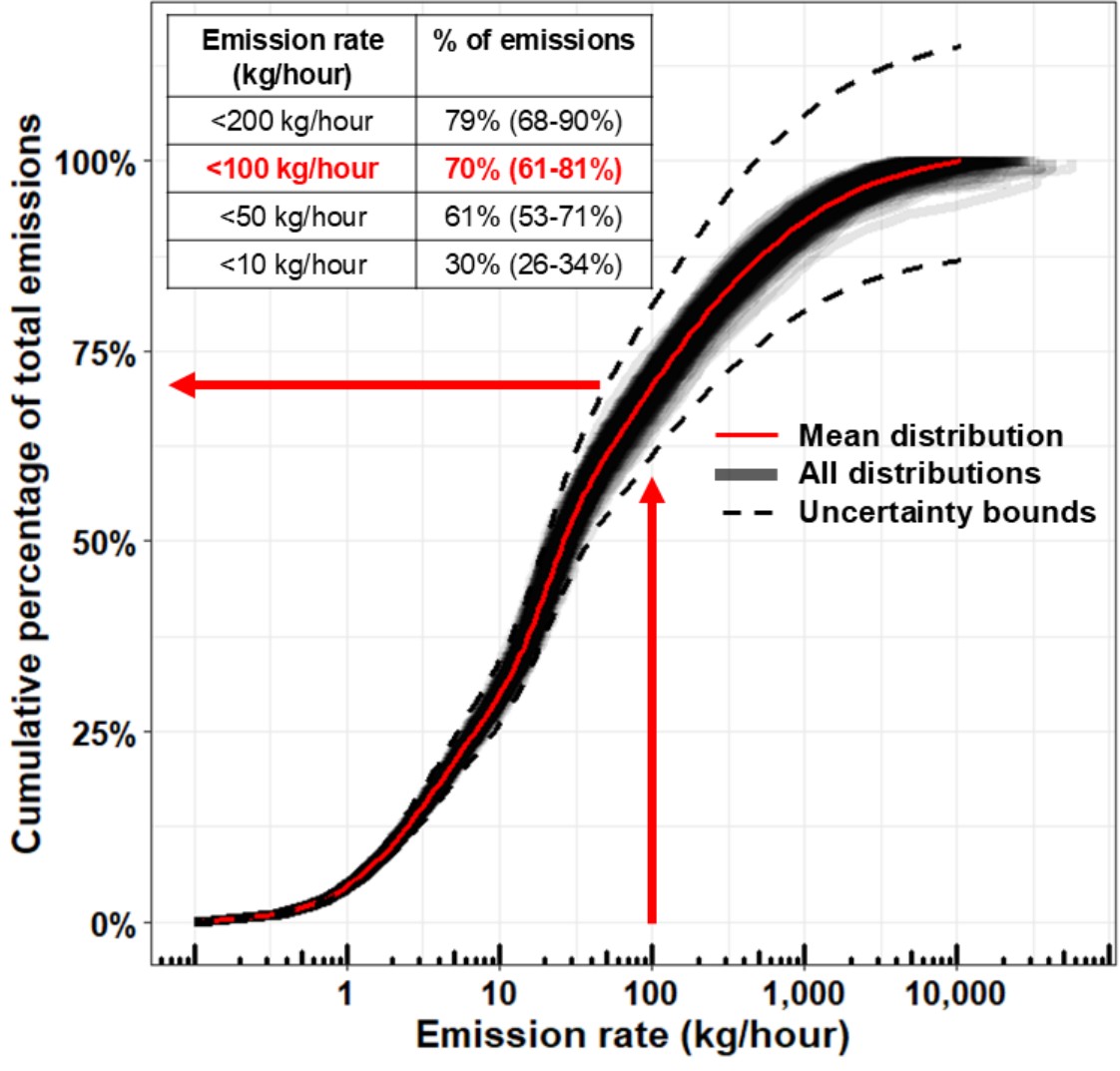


**Figure 3:** Results from 500 estimated facility-level emission distributions showing the cumulative percentages of
total methane emissions contributed from facilities emitting below methane emission rate thresholds. For example,
facilities emitting <100 kg/hr account for 70% (61-81%) of total methane emissions. The inset table in the upper left
displays the total percentage of methane emissions contributed from several discrete emission rate thresholds with
95% confidence intervals shown in parenthesis.


The distribution for our national-level methane emissions follows an S-shaped curve, noting that the x-axis

(i.e., facility-level methane emission rates) is presented in the $\log_{10}$ scale. From 0.1 to 1 kg/hr, we observe a plateau
in the distribution curve indicating that increasing emission rates within this range do not significantly increase the
percentage contribution to total regional emissions (Fig. 3), similar to the findings in Ravikumar et al. (2019). From 1
to 100 kg/hr, we see a sharper rise in the emission distribution, indicating that increasing emission rates at this range
lead to a more substantial contribution to total methane emissions, and account for 68% (60 – 75%) of total methane

emissions (Fig. 3, Table S4). Above an emission rate threshold of 100 kg/hr, we see an exponential decline in the percentage contributions of total emission with increasing emission rates, with total methane emissions in this range amounting to 28% (18 – 37%) of the total oil/gas emissions. Facilities emitting at the 1-10 kg/hr and 100-1,000 kg/hr ranges contribute a similar cumulative percentage at 26% (23 - 29%) and 22% (18 - 26%) respectively. Similar percentage contributions are also observed between the 0.1-1 kg/hr and >1,000 kg/hr ranges at 4.5% (4.0 - 5.1%) and 6.1% (2.6 - 13%) respectively. Overall, we find that the highest contribution to total national CONUS methane emissions occurs from facilities emitting in the 10-100 kg/hr range at 42% (37 - 46%). In terms of facility counts, from the 673,940 total active oil/gas facilities we estimate in the CONUS for 2021, we estimate that essentially all (i.e., ~99.9%) of these facilities emit methane below 100 kg/hr.

Our facility-level model estimates total methane emissions from US upstream/midstream oil/gas emissions for 2021 to be 14.6 (12.7 - 16.8) Tg/yr, or 1,668,000 (1,453,000 – 1,921,000) kg/hr (Fig. 4), which corresponds to a gross gas production normalized loss rate of 2.4%, assuming a uniform 80% methane content in natural gas across oil/gas producing regions in the CONUS. This national emission total of 14.6 (12.7 - 16.8) Tg/yr is more than double the EPA Greenhouse Gas Inventory Report for natural gas and petroleum systems in 2021, excluding post-meter and distribution methane emissions (Inventory of U.S. Greenhouse Gas Emissions and Sinks, 2024). We compare our total national estimates to previous estimates by seven studies that predominantly utilize satellite-based remote-sensing platforms such as GOSAT and TROPOMI inversions (Lu et al., 2022, 2023; Maasakkers et al., 2021; Shen et al., 2022; Worden et al., 2022) except for Alvarez et al. (2018) and Omara et al. (2024) who developed unique facility-based modeling approaches using empirical measurement data collected from multiple oil/gas basins in the CONUS (Fig. 4). Our estimate of national methane emissions overlaps with six out of seven other national estimates of oil/gas methane emissions for the US, with a combined average of 13.1 (ranging from 11.1 - 15.7) Tg/yr. We do not estimate methane emissions from gathering/transmission/distribution pipelines, post-meter emissions, abandoned oil and gas wells, and refineries due to the scarcity of measurement-based data for these sources. Total methane emissions from these sources emit ~2 Tg/year of methane emissions based on other studies (Williams et al., 2021; Alvarez et al., 2018; Omara et al., 2024; Weller et al., 2020; Inventory of U.S. Greenhouse Gas Emissions and Sinks, 2024). Overall, our total national estimate of CONUS methane emissions for 2021 shows

good agreement with multiple independent and recent measurement-based estimates.

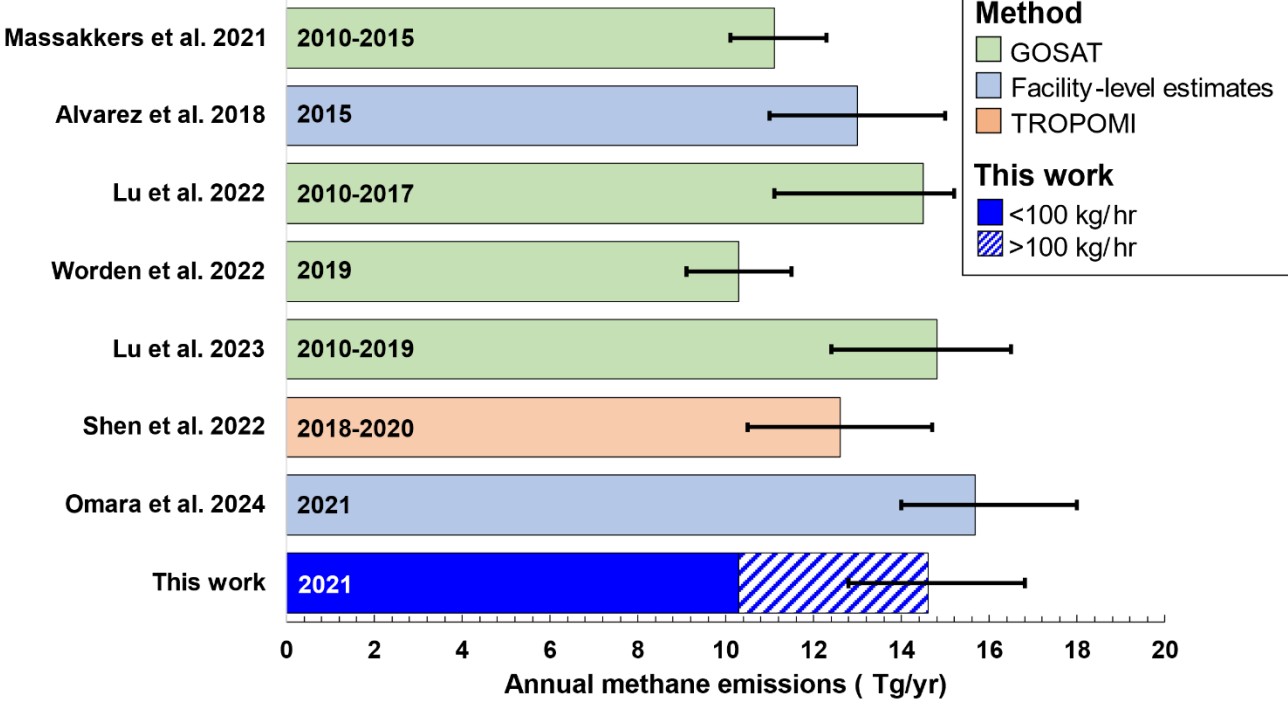

**Figure 4:** Comparison of total CONUS oil/gas emissions for 2021 from this facility-level measurement-based
inventory compared to empirical estimates from other studies. Bars are colored according to the methodology used
to derive the total national estimates, and the years within the bars represent the corresponding time periods for the
estimates. Black inset lines represent 95% confidence intervals. Our total estimates for "This work" do not include
emissions from other oil/gas methane sources such as abandoned oil and gas wells,
transmission/gathering/distribution pipelines, post-meter emissions, and refineries. Emission estimates from Omara
et al. (2024) do not include methane emissions from abandoned oil and gas wells. We assume that the remote
sensing estimates (i.e., GOSAT and TROPOMI) include all oil/gas methane sources, including downstream
emissions.

**3.2 Distribution of emission rates at the basin-level scale**

        Among the top nine emitting oil/gas basins in the CONUS, we observe variations among the different

basins in terms of the methane emission distributions, especially at higher emission rate thresholds (Fig. 5). The

majority of the top nine emitting oil/gas basins in Fig. 5 show higher percentage contributions from facilities

emitting <100 kg/hr when compared to our national estimate of 70% (61 – 81%) (Fig. 3). These percentage

contributions vary from ~80% in the Permian, Appalachian, and Eagle Ford basins, up to ~90% in the oil-dominant

San Joaquin basin. Only the Anadarko and Bakken basins have notably lower contributions to total emissions at the

100 kg/hr threshold at ~60% compared to the national level, which is still a significant majority of total methane

emissions. Despite these variations, our facility-level model estimates that the majority of total national oil/gas

emissions are consistently contributed from facilities emitting <100 kg/hr for the top nine emitting basins.

443   Our estimated facility-level emission distributions for the top nine emitting oil/gas basins all follow an S-

444 shaped curve (Fig. 5) like the national distribution (Fig. 3), albeit with certain variations. For all basins, the initial

445 plateau in the emissions distribution curves ends at around 1 kg/hr before beginning to rise more steeply. For the

446 Appalachian and San Joaquin basins, the second plateau is at the 20-50 kg/hr emission rate threshold (Fig. 5). For

447 the remaining basins, the rise in the emission distribution curves plateaus gradually, indicating a more consistent

448 relationship of emission rate thresholds to their contribution to total emissions. The variability displayed among the

449 500 basin-level simulations differs among the oil/gas basins, with less spread in the 500 estimated methane

450 emissions distributions for the Appalachian, Anadarko, and Permian basins compared to the Uinta, Denver-

451 Julesburg, and San Joaquin basins (Fig. 5 and Fig. S6). These variations are likely caused in part by the overall total

452 basin-level methane emissions, where an extremely high estimated methane emission rate would have a greater

453 impact on the percentage contribution to the total for basins with lower overall emissions (e.g., the apparent outliers

454 for the Greater Green River and Bakken basins in Fig. 5). We discuss below other plausible causes for basin-to-

455 basin variability in the estimated methane emission distributions.

456   In terms of total methane emissions, the top two emitting oil/gas basins are the Permian and Appalachian,

457 which collectively account for 5.2 (4.4 – 6.3) Tg/year (Fig. S1) or 37% of total upstream and midstream oil/gas

458 methane emissions. This exceeds the cumulative contribution from the other seven highest emitting oil/gas basins

459 which collectively account for 3.7 (2.9 – 5.0) Tg/yr. Notably, we find that the highest emissions in the CONUS

460 occur from regions outside of any basin boundary 4.3 (1.2 – 6.3) Tg/year. Our estimates for basin-level total

461 emissions also show good agreement with remote-sensing satellite-based observations (Fig. S1), except for the

462 Appalachian, Bakken, Greater Green River, and Denver-Julesburg basins where our results are consistently more

463 than double those from the remote-sensing studies that used a prior-emission based inversion result (Lu et al., 2023;

464 Shen et al., 2022). These four basins are in regions with relatively low TROPOMI observation counts and densities

465 compared to other regions in the CONUS (Shen et al., 2022), in addition to other factors that could influence

466 satellite-based inversions such as the presence of many non-oil/gas sources such as coal, livestock, and landfills.

467 Overall, our estimates of total basin-level emissions are consistent with satellite-based observations.

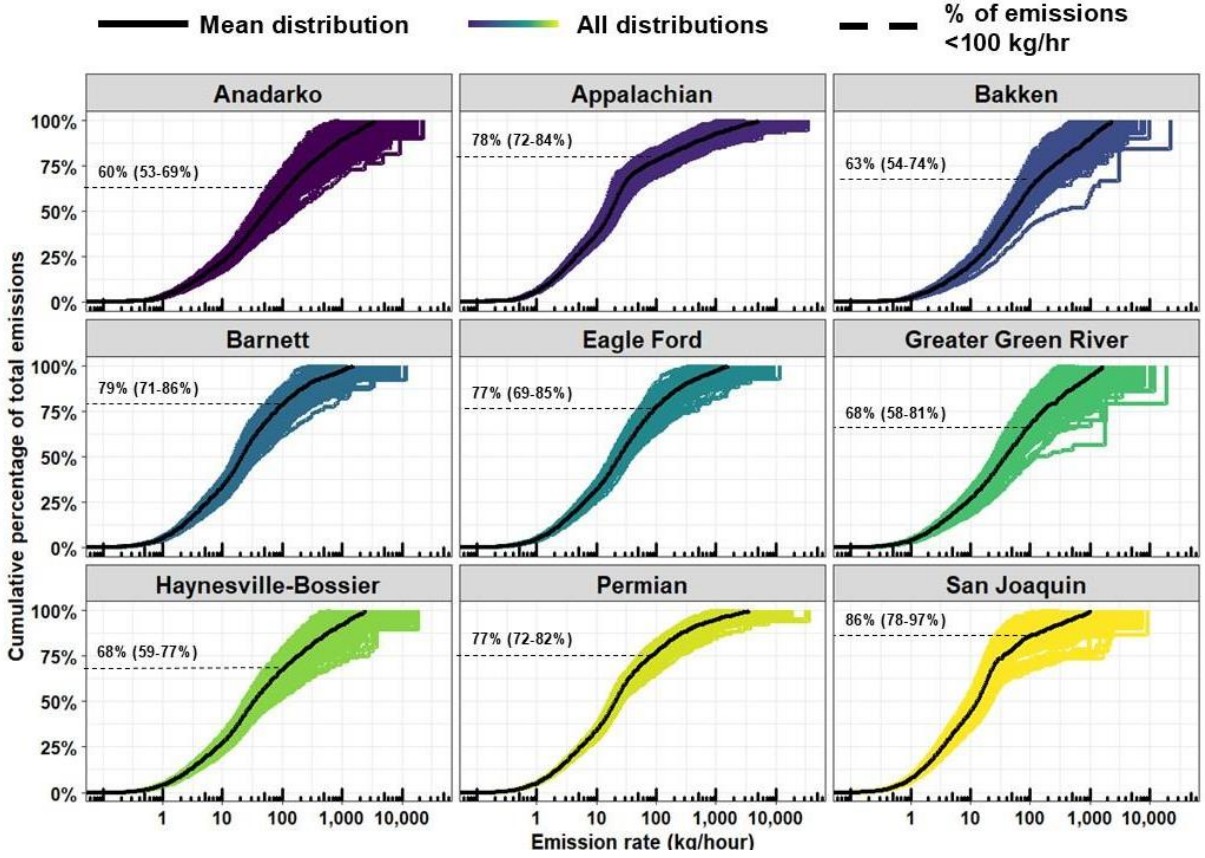

**Figure 5:** A) Results from 500 model simulations showing the cumulative methane emissions distribution curves for total upstream/midstream oil/gas methane emissions for the top nine emitting oil/gas basins in the CONUS for 2021. The model averages for each basin are shown in solid black lines. Inset dashed lines represent the percentage contributions of total emission from sources emitting <100 kg/hr. Emission distribution curves for the remaining eleven oil/gas basins in the CONUS are shown in Fig. S6, and a map of the spatial boundaries used for the different oil/gas basins is shown in Fig. S10.

## 3.3 Distribution of emission rates by facility category

We find significant variations in the methane emission rate distribution curves among the different facility categories (Fig. 6A). Over 50% of total methane emissions from low (i..e, <15 boe/day, or <0.13 kt of methane production per year and non-low production well sites, lit flares, and G&B compressor stations occur from facilities emitting <100 kg/hr (Fig. 6A). In contrast, only 17% (15-18%) of emissions from processing plants, 19% (18-20%) of emissions from T&S compressor stations, and 9% (7-12%) of emissions from unlit flares are contributed from emission sources <100 kg/hr. Similar variability is also observed at other emission rate thresholds, such as only 1% (0-2%) of total emissions for T&S compressor stations, unlit flares, and processing plants originating from facilities emitting at rates <10 kg/hr, compared to 50% (43-58%) from low producing well sites and 30% (24-35%) from non-low producing well sites (Fig. 6A). At higher emission rate thresholds, we find that 33% (20-45%) of total emissions from T&S compressors and processing plants are emitted from facilities <200 kg/hr, compared to 84% (68-93%)

from non-low producing well sites (>boe/day of combined oil and gas), 86% (83-88%) from VIIRS flare detections,
78% (70-86%) from G&B compressor stations, and essentially 100% of emissions from low producing well sites.
A breakdown of the 673,940 total facilities in our model has 541,970 as low-producing well sites, followed
by 121,824 non-low-production well sites, 4,431 G&B compressor stations 2,093 T&S compressor stations, 919
processing plants, and 3,153 total VIIRS flare detections. Of these 673,940 total facilities, 99.5% (99.4 – 99.6%)
emit methane at rates <100 kg/hr (Fig. S11), and in turn contribute 70% of total methane emissions (Fig. 3). Overall,
we estimate that 68% of total CONUS oil/gas methane emissions for 2021 come from production well sites, of
which 44% are from low-production well sites with combined oil/gas production <15 boe/day (i.e., <0.13 kt of
methane production per year), and the remaining 24% from non-low production well sites (i.e., >15 boe/day) (Fig.
6B). Midstream facilities contribute 29% of total methane emissions, with 13% from T&S compressors, 8% from
processing plants, 7% from G&B compressor stations. The remaining 4% from VIIRS flare detections are evenly
split with 2% each from lit and unlit flares respectively. Based on the population counts for each facility category
and their corresponding total methane emissions, the average methane emission rate per facility category is highest
for processing plants at 146 (115 – 283) kg/hr, followed by 106 (89 – 129) kg/hr for T&S compressor stations, 27
(25 - 29) kg/hr for G&B compressor stations, 3.3 (2.9 – 3.8) kg/hr for non-low producing well sites, and 1.3 (1.2 –
1.5) kg/hr for low producing well sites. For VIIRS flares detections, we find a large difference in average emissions
between lit flares at 11 (9.2 – 13) kg/hr and unlit flares at 205 (132 – 294) kg/hr.
Production well sites constitute the bulk of total methane emissions among the facility categories we
considered, with most of these emissions contributed from low production well sites. Overall, we find that 67-90%
of well site emissions originated from only 10% of national oil and gas production in 2021 (Fig. S7), highlighting a
disproportionately large fraction of emissions relative to production. In terms of individual well site level production
values, the same 67-90% of total cumulative methane emissions were contributed from well sites producing >50
boe/day (i.e., 0.43 kt of methane production per year) or lower. For well sites producing 15 boe/day (i.e., 0.13 kt of
methane production per year) or lower, which is the production threshold used to define a well site as being
marginally producing in previous work (Deighton et al., 2020; Omara et al., 2022), we find that these low producing
well sites accounted for 50-75% of total well site emissions, or 4.7-6.8 Tg/yr.


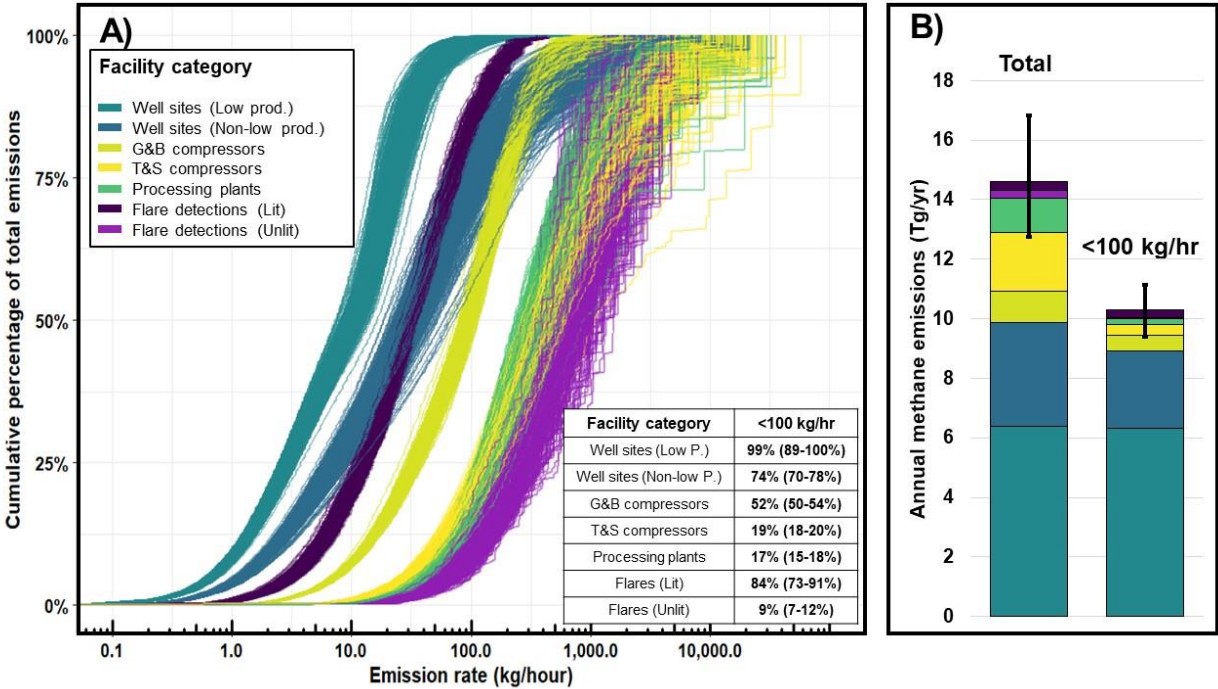


**Figure 6:** A) Results from an ensemble of 500 estimated methane emission distributions showing the percentage of total methane emissions among facility categories contributed from facilities emitting at rates below an emission rate threshold. The inset table on the bottom right displays the discrete percentage contributions to total methane emissions contributed from facilities emitting <100 kg/hr. B) Breakdown of total annual methane emissions contributed from all emitting facility categories and those emitting at rates <100 kg/hr.

## 3.4 Comparisons to aerial remote sensing studies

We perform comparisons of the percentage contributions of methane emissions from facilities emitting below discrete emission rate thresholds between seven aerial remote sensing campaigns across four distinct regions and our estimated facility-level results (Fig. 7). The aerial remote sensing technologies include data from Bridger GML measurements (Kunkel et al., 2023; Xia et al., 2024), MethaneAIR (Omara et al. 2024; Miller et al. 2023), and the results from Global Airborne Observatory and next-generation Airborne Visible/Infrared Imaging Spectrometer campaigns (Cusworth et al., 2022) which are also included in the aerial detections used by Sherwin et al. (2024). In comparing the percentage contributions to total emissions from low-emitting sources between our facility-level estimates and the aerial remote sensing campaigns, we find that emission contributions agree well across aerial remote sensing campaigns for the total percentage of methane emissions from facilities emitting, as seen in Fig. 7 for both less than 100 kg/hr and 200 kg/hr.

For the Bridger GML remote sensing campaigns (Kunkel et al., 2023; Xia et al., 2024), we find good agreement in the percentage of total emissions contributed from facilities emitting <200 and <100 kg/hr compared to our facility-level model estimates (Fig. 7). A comparison of continuous emissions distribution curves between our

facility-level emission distributions and two Bridger GML aerial remote sensing campaigns (Kunkel et al., 2023;
Xia et al., 2024) targeting four oil/gas basins is shown in Fig. S3. The Bridger GML aerial sampling platform has the
lowest LOD among the aerial campaigns we analyze in this work and a similar source resolution (i.e., 30 meters) to
our facility-level model (i.e., 50 meters), allowing for a more detailed comparison of continuous emission
distribution curves due to the higher number of detected methane sources at low emission rates provided by Bridger
GML surveys. We find close agreement between our facility-level methane emission distribution curves and the
observed emissions by Bridger GML in the four-basin aggregate provided by Xia et al. (2024) (Fig. S3A) which
includes Anadarko, Bakken, Eagle Ford and Permian basins (individual basin data are not currently available in Xia
et al. (2024)), as well as separately for the Permian remote sampling campaign (Fig. S3B) by Kunkel et al. (2023),
with the measured emissions from the Bridger GML surveys overlapping with our facility-level model simulations
throughout the continuous distribution of methane emission rates.

For the multiple aerial remote sensing campaigns performed by Cusworth et al. (2022), we generally find
good agreement with all of our estimates statistically overlapping for discrete emissions rate thresholds of <100
kg/hr and <200 kg/hr for the Permian and Uinta oil/gas basins (Fig. 7). For the San Joaquin and Denver-Julesburg
oil/gas basins, we see good agreement at the emission rate threshold of <200 kg/hr and at <100 kg/hr (i.e.
overlapping uncertainty bounds). For the Appalachian basin, we find broad agreement at both emission rate
thresholds of <100 kg/hr and <200 kg/hr, with our results consistently showing a 20-30% greater contribution from
emission sources below the discrete emission rate thresholds (Fig. 7). We find the closest agreement in the Permian
and Uinta oil/basins, where the differences in the average percentage contributions vary from -9% to +4% across the
three discrete emission rate thresholds of <100 and <200 kg/hr (Fig. 7). In Denver-Julesburg and Appalachian
basins, the differences are observed to be larger, compared to other basins, where the differences in average
percentage contributions across the discrete emission thresholds vary from -30% to +18%, however, they are within
our estimated uncertainty bounds. The detected point sources by Cusworth et al. (2022) in the Denver-Julesburg and
Appalachian basins contain many non-oil/gas point sources (Table S4), which may lead to additional uncertainty in
the comparisons for these basins since we use the relative proportions of point sources to subtract an estimated
contribution of non-oil/gas point sources from the TROPOMI estimates to provide a more direct comparison
between our estimates (since our study only focuses on upstream and midstream oil and gas sectors) and those of
Cusworth et al. (2022). Notably, the Appalachian basin contains the highest percentage contribution of non-oil/gas
point sources at 67% (Table S4). In contrast, we note that all of the detected point sources by Cusworth et al. (2022)
in the Permian and Uinta basins were attributed to oil/gas point sources (Table S4).

Our comparisons to the available flight results from MethaneAIR, which quantifies both total regional
methane emissions and high-emitting point sources >200 kg/hr from the same aerial platform (Chulakadabba et al.,
2023), show close agreement between our facility-level estimates and the available aerial campaigns in the Uinta
and Permian basins for facilities emitting <200 kg/hr (Fig. 7B). For the MethaneAIR flight in the Uinta basin, we
estimate that 92% (46 - 100%) of total oil/gas methane emissions are from sources emitting <200 kg/hr, compared to
88% from MethaneAIR (Fig. 7B). For the available flight in the Permian basin from MethaneAIR, we estimate total
contributions from sources emitting <200 kg/hr at 77% (59 – 90%) compared to the 71% estimated by MethaneAIR
(Fig. 7B).

Overall, our findings show that our facility-level estimates closely agree with the results from multiple

aerial remote sensing campaigns from different regions and using various measurement methods.

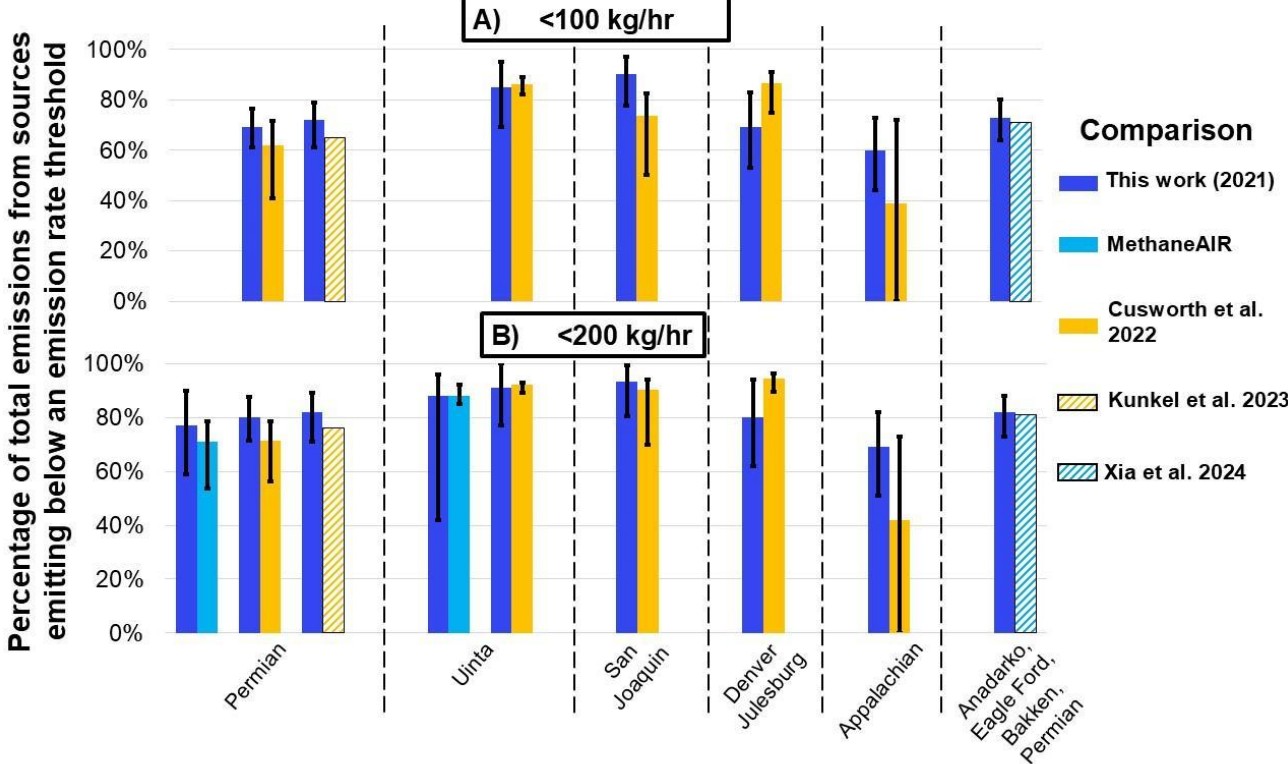


**Figure 7:** Comparisons of the cumulative percentage of oil/gas methane emissions from all oil/gas facilities emitting
A) <100 kg/hr, and B) <200 kg/hr, between our facility-level empirical emissions estimates and aerial remote
sensing campaigns. Bars are colored according to the study and grouped according to the target oil/gas basin(s). All
results from the facility-level simulations (i.e., this work) are constrained to the spatial boundaries of the aerial
campaigns for direct comparisons (note that for a given basin, spatial boundaries might be slightly different).
Uncertainty bars for the facility-level simulations are the 2.5th and 97.5th percentiles of 500 simulations. Maps of all
spatial boundaries used for comparisons are provided in Fig. S2. Comparisons to MethaneAIR are not performed at
the <100 kg/hr threshold because MethaneAIR detections are not available for point sources below this emission
rate threshold.

**4 Discussion**

Understanding how facilities with different magnitudes of emissions contribute to total regional emissions
has direct policy implications for methane quantification and mitigation, such as the selection of
measurement/screening methods with the appropriate detection sensitivities (Ravikumar et al., 2018). Our main
finding is that 70% of total oil/gas methane emissions from the upstream/midstream sectors come from facilities
emitting at rates <100 kg/hr, which is the emission rate threshold above which point source emissions are  referred
to as "super-emitting" oil/gas source by the EPA (Standards of Performance for New, Reconstructed, and Modified
Sources and Emissions Guidelines for Existing Sources: Oil and Natural Gas Sector Climate Review, 2024). While
detecting and mitigating emissions from super emitters are important (Cusworth et al., 2022; Duren et al., 2019;
Sherwin et al., 2024), our results underscore the need to account for oil/gas methane sources emitting at lower rates,
as the cumulative contribution of lower-emitting sites accounts for a large majority of emissions across US oil/gas
basins. Facility-level, measurement-based data collected in other countries present a similar story. From a sample of
sites (n=302) measured via Bridger GML remote sensing platform in British Columbia, Canada (Tyner and Johnson,
2021), roughly 60% of the total quantified oil/gas site-level emissions originate from sites emitting <32 kg/hr. In
Romania, a site-level measurement-based inventory (Stavropoulou et al., 2023) using 178 measurements finds that
oil production facilities emitting <100 kg/hr contribute 78% of total oil/gas methane emissions in the studied region.
In short, the high percentage contribution from lower-emitting (<100 kg/hr) oil/gas facilities that account for a large
majority of total emissions is not unique to the US and is likely present in other countries as well. A combination of
approaches that characterize entire emission distributions across populations of sites (i.e., not just focusing on
measuring super-emitters) and quantification of regional-level emissions is needed in other countries to quantify the
relative contributions of low-emitting sources that in aggregate can be significant sources of overall oil/gas methane
emissions. Most of our analysis centers around quantifying the percentage contributions of oil/gas methane sources
emitting below one discrete emission rate threshold (i.e., <100 kg/hr, per EPA's definition of a super-emitter). We
estimate that over 99% of the total oil/gas facilities that we analyze in this work emit below 100 kg/hr (Fig. S11),
which in turn contribute 70% (61 – 81%) of total methane emissions (Fig. 3). The emission rate threshold of 100
kg/hr is relevant to US policy decisions (EPA's Final Rule for Oil and Natural Gas Operations Will Sharply Reduce
Methane and Other Harmful Pollution., 2024), but we also illustrate the importance of a complete characterization of
emissions, which gains importance as newer methane monitoring technologies have different LODs. For example,
the effective LOD at high probabilities of detection for available point source imaging satellites of ~200 kg/hr
(Jacob et al., 2022) would only be able to quantify 21% (10-32%) of all oil/gas point sources in the CONUS, if the
full oil/gas sector was mapped in its entirety, based on our facility-level results. When considering the relationship
of facility-level emission rates to total cumulative methane emissions, we find that oil/gas methane emissions in the
CONUS are dominated by many low-emitting facilities, which relates directly to methane measurement
technologies.
Point source-focused remote sensing platforms offer the advantage of rapidly surveying large areas (i.e.,
100's-1,000's km$^2$) which facilitates the detection and quantification of high-emitting point sources (Cusworth et al.,
2022; Duren et al., 2019; Sherwin et al., 2024). In contrast, logistical constraints often limit the sample sizes for
ground-based vehicle sampling platforms, however, these limitations can be overcome with stratified random,
representative sampling and statistical analysis approaches like this work. Ground-based measurement platforms
provide much lower LODs (i.e., <1 kg/hr) when compared to remote sensing platforms, which are necessary to
quantify emissions from the large number of small methane sources we find that contribute roughly three-quarters of
total regional oil/gas emissions in the CONUS and will only improve as additional ground-based measurements are
gathered. Overall, our main findings highlight the importance of methods that can rapidly locate the small number of
high-emitting point sources we estimate, but our findings emphasize the need to account for the disproportionately
large majority percentage of total regional oil/gas emissions that are emitted from smaller diffuse methane sources.

When extrapolating our facility-level model results to the basin-level we see variations among the emission

distribution curves for different oil/gas basins, but still find that most methane emissions come from facilities
emitting <100 kg/hr. The variations in the emission distribution curves for different basins are driven by many
factors, such as the: production characteristics, number and density of facilities, different types and relative counts of
facility categories, the availability of empirical measurement data used to model emissions, and the total oil/gas
methane emissions (i.e., the denominator). For example, the Appalachian basin is dominated by a high number of
older low-production well sites (Deighton et al., 2020; Riddick et al., 2019; Enverus, 2024) with fewer midstream
facilities such as processing plants and G&B compressors, which contrasts with the Bakken basin where we find a
high number of midstream facilities, high-producing well sites, and VIIRS flare detections (Elvidge et al., 2015;
Enverus, 2024). When comparing the emissions distribution curves for the Bakken and Appalachian basins (Fig. 5),
we observe higher contributions from lower-emitting facilities for the Appalachian compared to the Bakken. An
example of differences in basin-level production is shown in Fig. S4 and Fig. S5, where we see variable profiles
among the different oil and gas-producing basins in terms of well site production characteristics, which are the main
source of total methane emissions in this work (Fig. 6). We also observe the influence of total basin-level emissions
on the variability among our emission distribution curves, where large emitting sources in the San Joaquin basin can
lead to high variability among the estimated emission distribution curves compared to the Permian basin which has
roughly ten times the total emissions compared to the San Joaquin (Fig. 5). We note that a direct comparison of our
model results with aerial remote sensing methods may be limited, in part, by methodological differences in methane
quantification approaches (and underlying uncertainties). The remote sensing observations assessed here as
snapshots may capture facility-level emission distributions that are not well represented in annually averaged
methane emissions distributions, as we estimate here. Nevertheless, we find broad agreement with these independent
aerial remote sensing estimates at the basin scale and across smaller spatial domains, as discussed. Ultimately, as
many characteristics will influence methane emissions distribution curves among different oil/gas producing regions
in the CONUS, mitigation strategies will need to be structured accordingly to the region they are targeting.

Our results find that over half of cumulative methane emissions from three different facility categories

come from facilities emitting <100 kg/hr, including methane emissions from lit and unlit flares. We show how the
large contributions from small methane sources to total regional emissions are not unique to any one facility
category, but it is important to contextualize our emission distribution curves with the corresponding total regional
emissions. Our facility-level estimates find that the main source of oil/gas methane emissions in the CONUS are
oil/gas production well sites, of which the low production category is responsible for 44% (39 – 49%) of the total
estimated oil/gas methane emissions in the CONUS in 2021. Low-producing well sites, also known as "marginal
wells", have been shown in previous work to be a significant source of methane emissions, especially relative to
their contribution to overall oil/gas production (Deighton et al., 2020; Omara et al., 2022). Omara et al. (2022) found
that marginal wells contributed anywhere from 37%-75% of total methane emissions from production well sites,
which is like our estimates (i.e., 50-75%). Despite low production well sites having a lower mean emission rate
compared to other facility categories, the large facility counts result in significant aggregate total emissions of
methane. This implies that detection and mitigation strategies to reduce methane emissions from these and other
low-emitting oil and gas infrastructure (e.g., abandoned oil/gas wells) would require alternative mitigation and
detection approaches compared to those for the small number of super-emitting emission sources. For detection,
measurement methods that can measure emission rates between 0.1-100 kg/hr are required, since this range makes
up 70% of total methane emissions (Figure 3 and Table S1) as modeled herein. In terms of methane mitigation
policy, financial incentives, like the USD 4.7 billion from the Biden Bipartisan Infrastructure Law for abandoned
wells, could be used to prioritize the repair of old and leak-prone production well sites, as these low-producing well
sites only account for a small fraction (i.e., 5.6% in 2019) of total oil/gas production (Omara et al., 2022).
We see good agreement between our facility-level results and a majority of aerial remote sensing studies, which
are expected to capture a wide range of high-emitting facilities in a survey region. For example, when comparing
our model results to Kunkel et al. (2023) and Xia et al. (2024) we find that our estimated methane emissions closely
match the distribution of methane emissions measured in Bridger GML surveys (Fig. S3). We also find good
agreement to satellite remote sensing estimates of emissions, such as our basin-level (Fig. S1) and national-level
comparison to satellite inversions (Fig. 3), and other aerial remote sensing study regions (Table S2). Our
comparisons of the contributions of low-emitting sources below discrete emission rate thresholds also agree closely
with recent MethaneAIR, Kairos Aerospace, GAO, and AVIRIS-NG aerial surveys, whose results also highlight the
importance of small methane sources to overall oil/gas methane emissions. Recently, Sherwin et al. (2024)
suggested that a majority of total emissions originate from a small fraction of high-emitting sites. Notably, most of
the aerial measurements that are used in Sherwin et al. (2024) are obtained from the Cusworth et al. (2022) study,
with which we see good agreement (Fig. 7). Sherwin et al. (2024) perform an alternative analysis than Cusworth et
al. (2022) for aerially measured sources with <3 overpasses and assume that sources with one or two overpasses
emit at their observed intermittency of 100%, 50%, or 0% of the time. This difference in analytical approaches
produces higher contributions from aerial emissions in Sherwin et al. (2024) by 31% on average for seven aerial
campaigns compared to Cusworth et al. (2022) (Table S7), which uses a resampling approach described earlier in
the Methods Section 2.4.  In addition, emissions from Sherwin et al. (2024) that are below aerial detection limits are
estimated using a combination of an equipment-level bottom-up model presented in Rutherford et al. (2021) for
production well sites, and emission factors from the U.S. Greenhouse Gas Inventory (Inventory of U.S. Greenhouse
Gas Emissions and Sinks, 2024) for midstream facilities, which produces 52% lower emissions on average for seven
aerial campaigns (Table S7). Therefore, the aerially measured emissions in Sherwin et al. (2024) are higher and the
emissions below aerial detection limits are lower which leads to a higher contribution to total methane emissions
from high-emitting facilities (Table S7). Ultimately, the broad agreement we find across multiple disparate
measurement techniques and platforms across Bridger GML aerial campaigns (Kunkel et al., 2023; Xia et al., 2024),
MethaneAIR measurements (MethaneAIR L4 Area Sources 2021 | Earth Engine Data Catalog, 2024; Omara et al.,
2024), and the multiple surveyed regions presented in Cusworth et al. (2022), altogether provide collective evidence
about the large contribution of smaller emission sources to total regional emissions.
Given the variability in methane detection technologies, a range of approaches can be taken to estimate methane
emission rate distributions, each providing unique advantages and disadvantages. MethaneAIR provides a novel
remote sensing approach where high-emitting point sources, distributed area sources and total regional emissions are
quantified using the same aerial platform, providing the ability to directly measure high-emitting point source and
diffuse area contributions to total regional estimates. In the work by Xia et al. (2024) they combine measurements
from Bridger GML across four oil/gas basins and use component-level simulations to account for facilities emitting
below the 3 kg/hr LOD of Bridger GML. Other approaches also exist, such as Cusworth et al. (2022) who combine
TROPOMI inversions to estimate total regional methane emissions with point source emissions quantified from their
aerial detection platforms (i.e., GAO, AVIRIS-NG). Similarly, Sherwin et al. (2024) combine point source emissions
measured via aerial remote sensing with site/facility-level emission rates estimates calculated from a combination of
an equipment-level bottom-up model for production well sites (Rutherford et al., 2021) and emission factors from the
2023 GHGI for midstream facilities (Inventory of U.S. Greenhouse Gas Emissions and Sinks, 2024) for facilities
emitting below aerial detection limits. Remote sensing studies have key advantages over ground-based sampling
platforms, such as rapidly surveying wide areas and capturing higher-emitting point sources, but have variable LODs
depending on the target region, topography, measurement technology, presence of co-located non-oil/gas methane
sources (i.e., source attribution), weather conditions, infrastructure density, and infrastructure type(s). These variables
pose additional challenges when quantifying the contributions from facilities emitting above/below specific emission
rate thresholds, which are critical information to inform mitigation policy. Assessing performance, tracking mitigation,
and accurate reporting requires building a comprehensive picture of emissions by characterizing all emitters big and
small, and reconciling with total basin/sub-basin level emissions. Ultimately, the key seems to be merging the best
data from both approaches to build a hybrid inventory, ideally using a multi-tiered system with multiple methods that
span a range of LODs that allow for gathering empirical measurements from facilities emitting at all parts of the
methane emission distribution curve. Our study is a step in that direction considering measurement-based data while
presenting a robust comparison with available independent remote sensing measurements. At the same time, large-
area aggregate emissions data obtained from wide-area remote sensing mapping or mass balance surveys can better
constrain total regional emissions (e.g. Cusworth et al. 2022; Omara et al. 2024) towards a more empirically robust
denominator in characterizing the relative contributions of small emission and high emission sources to total
emissions.
We show that our facility-level emission models produce national- and basin-level methane emissions estimates
that are in good agreement with other independent measurement-based studies. However, we note the following
limitations/biases that could be improved with future data collection efforts. The empirical measurements that we

use in our model are representative of the year and time they were measured (i.e., 2010-2020), meaning that they would not reflect any updates in regulatory practices or changes in facility operational and emission management practices. In addition, there are variations in the number of production well site empirical measurements among oil/gas basins (Table S3) although a sensitivity analysis shows that excluding data from individual oil/gas basins does not significantly impact our results (Fig. S9). Furthermore, there are several oil/gas methane emission sources that we do not account for in our estimates, which include: gathering/transmission/distribution pipelines, oil refining and transportation, abandoned oil/gas wells, offshore oil/gas infrastructure, post-meter sources, and oil/gas distribution infrastructure in urban areas. For some sources omitted in this work such as abandoned oil/gas wells, their inclusion would likely lead to a higher contribution from low-emitting facilities, since the highest recorded emission rate from an abandoned oil/gas well is 76 kg/hr (Riddick et al., 2024). For others such as oil refineries, their inclusion would likely lead to a lower contribution from small methane sources given their low facility counts and high per-site emissions (Duren et al., 2019). Despite their omissions, total methane emissions from these sources are currently estimated to account for 5-10% (Alvarez et al., 2018; Riddick et al., 2024; Inventory of U.S. Greenhouse Gas Emissions and Sinks, 2024; Williams et al., 2021) of total oil/gas sectoral emissions. Our estimates also utilize empirically measured emission rates from ground-based sampling platforms which are limited in number, especially in the case of processing plants (n=20) and T&S compressor stations (n=50) (Table S2). The empirical data used in our analysis includes a smaller sample of super-emitting facilities relative to those captured by remote sensing platforms (Duren et al., 2019; Sherwin et al., 2024), but our use of production-normalized loss rates and lognormal distributions to estimate facility-level methane emission rates anticipates and accounts for the possibility of finding low-probability, high-magnitude emissions that occur at rates beyond those that appear in our dataset of empirical observations. For example, our highest empirical emission rate is 1,360 kg/hr for a T&S compressor station, whereas our maximum estimated facility-level emission rate across all 500 facility-level emission distribution curves averages 7,500 kg/hr (3,000 - 21,000 kg/hr). Finally, we include a small number (i.e., 5% of total empirical data used in the model) of measurements for production well-sites gathered using ground-based component/source-level sampling methods from two studies (Deighton et al., 2020; Riddick et al., 2019). All measurements from these two studies targeted the lowest production cohort of production well sites and exhibited statistically lower emission rates than those gathered using facility-level ground-based methods for the same well site production cohort, meaning that any bias introduced by the inclusion of these measurements would lead towards the underestimation of total emissions and/or the percentage contributions from low-emitting sources. Despite these limitations, we have shown that our results are broadly in agreement with satellite- and aerial-based remote sensing studies at national/basin/local scales, and other facility-level estimates.

Going forward, several approaches can be used to better understand the percentage contributions from facilities emitting at different leak rate thresholds, and ultimately improve our understanding of oil/gas methane emissions in the CONUS and around the world. A combination of multiple satellite and aerial remote sensing approaches and synthesis of their data by bringing in point source detections at multiple thresholds at the same time characterizing total regional emissions as demonstrated using a compilation of multi-scale measurements seems a viable pathway towards building a more complete picture of the overall methane emissions. Combining aerial and satellite remote

sensing measurements with ground-based site/facility-level estimates presents itself as an effective next step, as implemented/suggested by prior studies (Allen, 2014; Alvarez et al., 2018). Aerial or satellite remote sensing platforms focused on point source detection offer the ability to rapidly locate the small number of the highest emitting facilities that contribute a disproportionate fraction of emissions, offering valuable data on specific facility locations that allow for rapid mitigation. However, more direct observational approaches are needed to acquire total emissions data which according to this study is dominated by small-emitting sources that are undetected by high-emitting point source detection systems. Facility-level population-based approaches can account for the lower-emitting facilities that contribute the most total oil/gas methane emissions, which is needed for accurate emission reporting and understanding the contributions of total emissions above/below emission rate thresholds. The ground-based estimates can be further constrained by large-area aggregated emission quantification provided by regional remote sensing or mass balance mapping approaches (Shen et al., 2022; Omara et al., 2024; Jacob et al., 2022) towards producing a more robust overall emission quantification.

**5 Conclusions**

In conclusion, our work highlights several key aspects of oil/gas methane emission rate distribution curves in the CONUS for 2021, which include:

1. A large majority (70%) of total national continental oil/gas methane emissions in the US originate from lower-emitting facilities (<100 kg/hr).

2. Emission rate distributions vary among different oil/gas basins, but among the top nine producing basins we consistently find that most methane emissions (60%-86%) originate from oil/gas facilities emitting at rates <100 kg/hr.

3. Production well sites were found to be responsible for 70% of regional oil/gas methane emissions, from which the sites that accounted for only 10% of national oil and gas production in 2021, disproportionately accounted for 67-90% of the total well site emissions.

4. Our results were consistently found to be in close agreement with those from independent aerial/satellite remote sensing estimates, both in comparing contributions from discrete emission rate thresholds and continuous emissions distribution curves, which emphasize the importance of the large majority contribution of small-emitting methane sources to total oil/gas methane emissions.

Our results highlight, and quantify, the significant contributions of the large number of low-emitting oil/gas facilities to total regional/basin/local oil/gas methane emissions in the CONUS for 2021. In addition to the CONUS, the small oil/gas methane sources are likely a significant component of total regional emissions in other countries as well as recent data suggest from Romania and Canada (Stavropoulou et al., 2023; Tyner and Johnson, 2021) and would need to be further investigated to build a comprehensive assessment of small-emitting methane emissions and their relative contributions to total oil/gas methane emissions globally. This work emphasizes the need for multi-

scale approaches to quantify total regional oil/gas methane emissions; and at the same time characterize and account
for the large contribution from small emission sources dispersed across a wide area, in addition to incorporating data
on high-emitting point sources towards producing overall robust methane emission quantification.

**Data availability**
All 500 full emission rate distributions at the national level are available to download from Zenodo (link:
https://doi.org/10.5281/zenodo.13314532). All estimated methane emission rate distributions at the basin or small
target scale are available upon request. Empirical measurement data used in the estimation of the methane emission
distribution curves are available from the references listed in Table S2.

**Code availability**
R code used to create the methane emission distribution curves and figures is available upon reasonable request.

**Acknowledgements**
We acknowledge funding support from the Bezos Earth Fund. We would like to thank Jack Warren and Luis Guanter
for their valuable efforts in analyzing point source emissions from MethaneAIR aerial campaigns.

**Author contributions**
JPW and RG designed this study. JPW created the code used to produce all results, with inputs from MO, KM, DZA,
and AH. MethaneAIR analysis was provided by JB, MS, and SW. Multi-sensor airborne intercomparison was
performed by JPW and RG. JPW prepared the manuscript with input from all co-authors.

**Competing interests**
The authors declare that they have no conflict of interest.

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
