# Peer review of "Small emission sources in aggregate disproportionately account 1 for a large majority of total methane emissions from the US oil 2 and gas sector 3"

_EGUsphere, 2024_

## Referee Comment (RC1)

**Review of: *Small emission sources disproportionately account for a large majority of total methane emissions from the US oil and gas sector**

Authors: James P. Williams, Mark Omara, Anthony Himmelberger, Daniel Zavala-Araiza, Katlyn MacKay, Joshua Benmergui, Maryann Sargent, Steven C. Wofsy, Steven P. Hamburg, Ritesh Gautam

*Summary*:
This work uses emission factors from ~20 published studies across ~9 regions to estimate national methane emissions from active mid- and up-stream oil/gas production facilities for 2021. Using infrastructure inventories (Enverus, OGIM database), regional emission rates were modelled and validated with airborne surveys.

The manuscript is well written and the subject is of suitable content for EGUsphere. The subject is timely as there is active discussion regarding how mitigation funding can most effectively be used to reduce fugitive emissions from O&G. The figures are well designed and informative.

I have two main hesitations that together question the novelty of this work and the contributions that it provides. First, the chosen methodology, which is complicated and I am not convinced contributes to the authors results, discussion, or conclusions (see general comment 1). Second, the close similarity of this work prior work from this group (see Omara et al. 2022; 2024) questions the novelty of this manuscript. Specifically, the aggregation of emission factors is already published in Omara et al. 2018, 2022, & 2024. Scaling from emission factors to national budgets using Enverus is repeated from Omara et al. 2022 & 2024. Lastly, comparison of national/regional/basin-level emissions to airborne studies was previously done in Omara et al. 2022 & 2024.

Given the incremental differences between this manuscript and others that this group has published, I recommend declining publication.

*General Comments:*

1. Methodology: What is the benefit of using a bootstrapping approach? Is the bootstrapping solely to provide confidence intervals, or is there an additional benefit?

    My criticism is that many of the same results and conclusions are achieved without this analysis or less complex approach. Conclusions 1 and 2 can be drawn solely from the prior EF distributions. Conclusions 3 and 4 require knowing the number of facility types and production rates (taken from Enverus, OGIM database) but also do not require the monte carlo bootstrapping. Same critique for sections 3.1, 3.2, and 3.3.

As an example, the main conclusion of the authors (Conclusion 1, L712) is that 72% (70% as stated in abstract, L22) of total emissions are from facilities that emit less than 100 kg/hr. This is in fact buried in the last table of the supplement, which states the prior emission distribution and shows that 72.7% emissions facilities are from these "small" emitters. The posterior is unchanged from the prior, which is good since MC bootstrapping in this approach shouldn't change the center value.

A MC bootstrapping technique may be more interesting if applied to randomly select which EF studies to include. For example, if 6 of the 11 studies of facility category "Well Sites" listed in Table S1 were randomly selected for each simulation, then we might assess dependence based on regional dependence of studies, sampling/analytical methodology, etc. Indeed, regional differences are maybe observed, e.g. loss rates of 0.90% for Appalachian and Greater Green River regions (Omara et al, 2018) compared to >4.5% for San Joaquin and San Juan regions, but the variance within the regional populations precludes saying these loss rates are different (based on a Tukey test). Could the Tukey test be run on the $log_{10}(loss \%)$, given that these appear to be lognormally distributed in Figure 1?

2.  What is the 95% CI for the total national CH4 emissions?
3.  Data Availability: Data should be made available in a publically accessible, reliable repository and linked, preferably, through a DOI per EGUsphere instructions.

    Ideally, I would also prefer to see a table or reference section in the supplementary that has direct links, references, etc to the data from other studies used in this manuscript. This would be the data references in Table S1, plus Lan et al. 2015.

*Specific Comments:*

| Line # | Comments |
|---|---|
| 62 | Would be useful to state what the LOD of Bridger GML is here. |
| 106 | "1,898 facility-level…" I am a bit confused since Table S1 only sums to 1866 observations. |
| 127 | "high-emitting intermittent are included" → "high-emitting intermittent sources are included" |
| Fig. 1 | There appears to be a linearly decreasing relationship between the loss % and production rates for well-sites (facility category 5-9). Is this real? Is there a reason to include this in the facility-level model? |

| | |
|---|---|
| 226 | "… gas flared for 2021 by Elvidge et al. (2016)… efficiencies from Plant et al. (2022)" Are these the correct references? It seems unlikely that Elvidge et al (2016) published gas flaring for 2021. |
| 254 | "…production well sites that we use in this work generally do not show significant…" → "… basin-to-basin, production well sites in …" |
| 352 | "… Ravikumar et al. (2019) From …" → "…Ravikumar et al. (2019). From…" |
| Fig 4 | What do the error bars represent? 95% CI? |
| 534 | "our results show the essentiality of expanding beyond solely on super-emitter mitigation". Some sort of grammatical correction needed. |
| 538-540 | It would be nice to provide the sample size of these studies. |
| Table S1 | Appears to be missing a reference to Lan et el. 2015.

There are several other references used by Omara 2018 not included in this study. (Goetz et al 2015, ERG 2018) |
| Table S2 | The total number of well sites for the Barnett basin is 32 wells less than the sum of the bins. I assume this is the 32 wells measured by Lan et al. (2015) that was not included in Table S1. |

---

## Author Comment (AC1)

**Author response to open comments**

**Open comment by Daniel Cusworth**

Williams et al. presents a very extensive summary of U.S. oil&gas emissions using a combination of bottom-up modeling and atmospheric observations. The breadth of the survey should be commended for bringing additional information to this important emission sector. I do have one comment to help clarify the study, especially as it relates to the study's title. The use of the wording "disproportionate" is not supported by the results of the study. In fact, the authors' conclusion in this manuscript is that the majority of emissions result from small emitters, and that small emitters represent the majority of infrastructure in oil&gas basins. Therefore, the aggregate emissions from small sources are essentially proportionate to their numbers. It would be clearer and more correct to strike the word "disproportionate" from the title. This seems particularly important given that one of the author's main points is that methane mitigation policy needs to address emissions from large numbers of smaller emitters in addition to a small number of super-emitters.

We thank Dr. Daniel Cusworth for taking the time to read through our pre-print and offer his insights on this paper.

We do agree that the word "disproportionately" as used in the current title may be misconstrued to indicate a disproportionate relationship between facility counts and total emissions. In the case of our work, when we use the term "disproportionate" we are referencing the large majority of cumulative emissions contributed from facilities emitting at relatively low emission rates (i.e., <100 kg/hr/facility). In addition, our work also finds a large majority of cumulative emissions from well sites (i.e., the facility category contributing 68% of total emissions from our estimates) that contribute a small percentage of overall production. To clarify these points, we have made some changes to the text in the main paper, including a slightly revised title and new figures in the SI. We hope that the following changes address the comments expressed by Dr. Cusworth.

- Revised title of the manuscript:
  - [title] *"Small emission sources in aggregate disproportionately account for a large majority of total methane emissions from the US oil and gas sector"*

- New text in the Abstract/Results/Conclusions that highlights the disproportionately high emissions from well sites contributing a minority of overall production:

  - [page 1] *"We estimate that production well sites were responsible for 70% of regional oil/gas methane emissions, from which we find the well that accounted for only 10% of national oil and gas production in 2021, disproportionately accounted for 77% (72-81%) of the total well site emissions."*

  - [page 20] *"Production well sites constitute the bulk of total methane emissions among the facility categories we considered, with most of these emissions contributed from low production well sites. Overall, we find that 77% (72-81%) of well site emissions originated from only 10%*

*of national oil and gas production in 2021 (Fig. S7), highlighting a disproportionately large fraction of emissions relative to production. In terms of individual well site level production values, the same 77% (72-81%) of total cumulative methane emissions were contributed from well sites producing 0.43 kt/yr (0.43-0.45 kt/yr) or lower. For well sites producing 15 boe/day (i.e., 0.13 kt/yr) or lower, which is the production threshold used to define a well site as being marginally producing in previous work (Deighton et al., 2020; Omara et al., 2022), we find that these low producing well sites accounted for 65% (58-69%) of total well site emissions, or 6.4 Tg/yr (4.7-6.8 Tg/yr)."*

- *[page 31] "3. Production well sites were found to be responsible for 70% of regional oil/gas methane emissions, from which the sites that accounted for only 10% of national oil and gas production in 2021, disproportionately accounted for 77% (72-81%) of the total well site emissions."*

- A new figure in the SI that illustrates the relationship between well site production and cumulative emissions from well sites

[Figure]

- *"Figure S7: Results from 500 model simulations showing the cumulative methane emissions distribution curves for total well site oil/gas methane emission rates versus the percentage of cumulative combined oil and gas production. Results are ranked first by individual well-site*

*emission rates, and then by well-site combined oil and gas production. The inset table shows the specific percentages of total emissions contributed from production well sites for cumulative well site production values of 1%, 5%, 10%, and 20%. The red arrows correspond to the percentage of total well site emissions contributed from well sites cumulatively producing 10% of total CONUS oil and gas production in 2021."*

---

## Author Comment (AC2)

**Author response to reviewer comments**

**Anonymous Reviewer #1**

This work uses emission factors from ~20 published studies across ~9 regions to estimate national methane emissions from active mid- and up-stream oil/gas production facilities for 2021. Using infrastructure inventories (Enverus, OGIM database), regional emission rates were modelled and validated with airborne surveys.

The manuscript is well written and the subject is of suitable content for EGUsphere. The subject is timely as there is active discussion regarding how mitigation funding can most effectively be used to reduce fugitive emissions from O&G. The figures are well designed and informative.

We thank the reviewer for the valuable comments and edits, and we hope the following responses address their concerns.

All of the page references in the responses below reference the attached manuscript with tracked changes included. Text in *"bold blue italics"* references prior text from the manuscript, and text in *"bold red italics"* references new added text.

I have two main hesitations that together question the novelty of this work and the contributions that it provides. First, the chosen methodology, which is complicated and I am not convinced contributes to the authors results, discussion, or conclusions (see general comment 1). Second, the close similarity of this work prior work from this group (see Omara et al. 2022; 2024) questions the novelty of this manuscript. Specifically, the aggregation of emission factors is already published in Omara et al. 2018, 2022, & 2024. Scaling from emission factors to national budgets using Enverus is repeated from Omara et al. 2022 & 2024. Lastly, comparison of national/regional/basin-level emissions to airborne studies was previously done in Omara et al. 2022 & 2024.

Our understanding is that the reviewer is suggesting that Conclusions 1 and 2 [Section 5 – page 25] in the paper can be reached solely from the empirical measurements and emission factors without any extrapolation/modeling of distributions, which we respectfully disagree with.

Multiplying an average methane emission factor by the number of facilities can produce a rough estimate of total methane emissions, but is not a suitable approach for characterizing facility-level methane emission distributions, which must account for the stochasticity in facility-level methane emissions profiles and related uncertainties (for references that discuss this stochasticity: https://www.nature.com/articles/ncomms14012, https://pubs.acs.org/doi/full/10.1021/acs.est.6b04303, https://pubs.acs.org/doi/full/10.1021/acs.est.8b03535) Developing robust methods for characterizing such distributions at the basin- and national-scale is the focus of this work. While we do present estimates of total emissions estimates at the national/basin/aerial spatial scale, these are a by-product of our methodology and not the main findings, which are the detailed distributions of individual facility-level emission rate and the large majority contribution of total emissions linked to an aggregate of smaller emitting sources (i.e., the distributions presented in Figures 3, 5, and 6).

If, for example, an EPA GHGI emission factor (e.g., average methane emission rate per facility) and the associated confidence bounds (e.g., standard deviation of the mean) are applied to each individual facility

to provide an independent emission rate, and this is repeated for all facilities in the CONUS, this simplified approach would not produce an accurate distribution of emission rates because a representative methane emission factor would still need to account for (i) facilities that may be non-emitting at any one time, (ii) the fact that different facility categories (including different production ranges of well sites) can emit at different rates at any one time, and (iii) the representativeness of facility-level empirical data (and inherent uncertainties in emissions quantification) when compared with the national population of facilities.

For these reasons, we believe a probabilistic modeling approach that accounts for these factors (and others) is essential to assessing emissions distributions and underpins the novel findings we present in this work. Moreover, the conclusions in terms of the specific emission rate thresholds and the aggregate emissions below those and their relative fractions to the total emissions across the US oil and gas upstream and midstream sectors as well as over each individual oil/gas basin has not been produced before based on empirically derived measurement-based analysis, which this study presents as a major step forward in our understanding the dynamics of oil/gas emissions and their source contributions which have important policy implications for measurement and mitigation, as we have discussed throughout the manuscript.

For the second comment mentioned by the reviewer regarding the methods in this work sharing similarities with previous studies (e.g. Omara et al. 2018,2022,2024), each of these previous studies had a different scope and presented different data outputs than our work. Our work differs from these previous studies by Omara et al. by i) estimating full methane emission distribution curves (i.e., not just total methane emissions) across multiple spatial scales ranging from the entire CONUS to the aerial remote sensing campaign survey regions, ii) presenting both the cumulative, and grouped, emission distribution curves for major oil/gas emitting facility categories, which allows for the clear distinction between emission distribution curves from different facility categories and a relative assessment of their total contributions to CONUS annual emissions, iii) presenting detailed comparisons to prior work (i.e., satellite and remote sensing studies) on the distribution of methane emissions across spatial scales and at different emission rate thresholds, which involved the additional analysis of data provided by other studies, and iv) a revised methodology that utilizes new ground-based facility-level measurement data and a separate approach for accounting for lit and unlit VIIRS flare detections. As we discuss in the Main Text, characterizing the full methane emissions distribution curves, that is, the contributions from individual facilities emitting below or above an emission rate threshold is crucially important for effective methane emissions mitigation. For example, any methane measurement platform with an established limit of detection and/or 90% probability of detection, could reference the emission distribution curves we present in our work, and determine a rough approximation of what percentage of total methane emissions their measurement method technique could capture. Our work represents the first comprehensive attempt to develop such an emissions distribution curve, using empirical measurements collected from ground-based measurement methods and robust probabilistic models to characterize the facility-level distributions for the full US upstream and midstream oil and gas methane sources. As part of this work, we estimate emissions from major upstream and midstream oil and gas methane sources, including well sites, natural gas gathering and transmission compressor stations, natural gas processing plants, and emissions from natural gas flaring facilities, accounting for methane emissions from both the lit and unlit flares.

- In order to improve the clarity of our methods, and to better illustrate differences between our methods/results and other studies, we have added the following text in the manuscript.
  - [page 6-7] *"We calculate annual methane emissions from all facility categories (i.e., six production bins of production well sites, T&S compressor stations, G&B compressor stations, and processing plants, and VIIRS flare detections) using a multi-step probabilistic modeling approach adapted from multiple studies (Omara et al., 2018, 2022; Plant et al., 2022) (Fig. 2). Briefly, for each individual facility and VIIRS flare detection in the CONUS for 2021, we*

*estimate an annually averaged methane emission rate using empirical measurement data, and consequently the cumulative distribution of methane emission rates from the aggregation of these individual emission rates. Each emission rate estimate is indexed according to the corresponding replicate (n=500), and we use these repetitions to determine uncertainty for the cumulative methane emission distribution curves. The detailed steps of this process for all facility categories and VIIRS flare detections are described below.*"

- Revised Figure 2 added:

[Figure]

*DRE = Destruction removal efficiency

- *"Figure 2: Flowchart describing the facility-level estimates, with steps colored according to the specific process and data being used. We note that methane emission rates for flares are calculated using a separate approach from that of production well sites and midstream facilities. Processing plants and T&S compressors are excluded from the determination of whether a facility is a top 5% emitter due to a lack of available empirical measurement data.*"

- New text added in the Methods concerning VIIRS flare detections

  - [pages 8-9] *"For all VIIRS flares detections, we use the total reported volumes of gas flared for 2021 from flares detected using the VIIRS instrument (Elvidge et al. 2016) multiplied by the observed flare destruction efficiencies and percentage of unlit flares from Plant et al. (2022) to calculate annual methane emission rates from this source. As previously stated, our empirical measurements are largely located outside of oil/gas basins where the majority of VIIRS flare detections are located (i.e. Permian, Eagle Ford, and Bakken), but we cannot discount the possibility that there are instances of double-counting flares measured via our ground-based*

*empirical data and those detected by VIIRS. For each VIIRS flare detection, we randomly determine whether it is an unlit or lit flare based on the basin-specific percentages of unlit flares reported by Plant et al. (2022). If a flare is determined to be lit, we use the corresponding basin-specific observed destruction removal efficiencies as reported by Plant et al. (2022) multiplied by the corresponding annual total volume of gas flared and convert to an emission rate. The basin-specific observed destruction removal efficiencies are estimated through a fitted normal distribution using the mean and standard deviations modeled from the 95% confidence intervals presented in Plant et al. (2022). If a flare is determined to be unlit, we use a destruction removal efficiency of 0%. For VIIRS flare detections located outside of the Bakken, Eagle Ford, and Permian basins, we used the total CONUS averaged flaring efficiencies destruction removal efficiencies of 95.2% (95% confidence interval: 94.3 – 95.9%) and percentage of unlit flares of 4.1% as reported by Plant et al. (2022).”*

*General Comments:*

1) Methodology: What is the benefit of using a bootstrapping approach? Is the bootstrapping solely to provide confidence intervals, or is there an additional benefit?

The bootstrapping in this work is used for developing a probability distribution of a given facility emitting below the method LOD (i.e., 0.1 kg/hr), or being a top 5% emitter (in some cases). It is one of several ways in which we incorporate different facets of uncertainty into the estimates. The benefit of utilizing a bootstrapping approach is to include uncertainty associated with the chance of a facility being above/below the method LOD and a top 5% within the modeled outputs, which is then reflected in the estimated emission rate distributions. We would also point to a previous response regarding the stochasticity of facility-level emission rates for oil/gas facilities.

- [Previous response] "Multiplying an average methane emission factor by the number of facilities can produce a rough estimate of total methane emissions, but is not a suitable approach for characterizing facility-level methane emission distributions, which must account for the stochasticity in facility-level methane emissions profiles and related uncertainties (for references that discuss this stochasticity: https://www.nature.com/articles/ncomms14012, https://pubs.acs.org/doi/full/10.1021/acs.est.6b04303, https://pubs.acs.org/doi/full/10.1021/acs.est.8b03535) Developing robust methods for characterizing such distributions at the basin- and national-scale is the focus of this work. While we do present estimates of total emissions estimates at the national/basin/aerial spatial scale, these are a by-product of our methodology and not the main findings, which are the detailed distributions of individual facility-level emission rate and the large majority contribution of total emissions linked to an aggregate of smaller emitting sources (i.e., the distributions presented in Figures 3, 5, and 6)."

- We have added some new text in the Methods that describes some of the reasoning behind these methods.
  - [page 8] *"Next, we remove the empirical measurements below the LOD and use bootstrapping with replacement (n=1,000) on the above LOD empirical measurements to determine the probability of an emitting facility being in the top 5% (i.e., 95th percentile or above of empirical measurement data) or bottom 95% (i.e., 95th percentile or below the empirical measurement data) of emitters, except for processing plants and T&S compressors which had too few measurements (n=20 and n=50 respectively) to distinguish between the top 5% and bottom 95% of emission or loss rates. This pseudo-random selection of a top 5% emitter within each facility category accounts for the functional definition of abnormally large emissions (i.e., super-emitters) that can be observed in all facility categories (including well sites in different production bins) (Zavala-Araiza et al. 2015, Brandt et al. 2016)."*

My criticism is that many of the same results and conclusions are achieved without this analysis or less complex approach. Conclusions 1 and 2 can be drawn solely from the prior EF distributions. Conclusions 3 and 4 require knowing the number of facility types and production rates (taken from Enverus, OGIM database) but also do not require the monte carlo bootstrapping. Same critique for sections 3.1, 3.2, and 3.3.

We have included the following response as an expansion to an earlier comment by the reviewer below which we believe addresses the main concerns:

- Multiplying an emission factor by the number of facilities would not be able to provide individual facility-level emissions, which is our focus, but rather an aggregated total of emissions without any information on how much methane is being emitted above/below a given emission rate threshold. While we do present estimates of total emissions estimates at the national/basin/aerial spatial scale, these are a by-product of our methodology and not the main findings, which are the detailed distributions of individual facility-level emission rate (i.e., the distributions presented in Figures 3, 5, and 6).

  If an emission factor (i.e., average emission rate) and the associated parameters (i.e., standard deviation of the mean) are applied to each individual facility to provide an independent emission rate, and this is repeated for all facilities in the CONUS which are then combined together, then this would form the base of our methodology. However, this approach would not produce an accurate distribution of emission rates because the emission factor would still need to account for facilities that are non-emitting, the fact that different facility categories (including different production ranges of well sites) emit at different rates, that the available empirical measurement data for well sites does not share the same production characteristics as the entire CONUS, and that the measurements used to derive this emission factor have inherent uncertainties. After accounting for these factors (and others) we begin to reconstruct the methodology used in our work, which we believe is essential to produce the findings we present.

As an example, the main conclusion of the authors (Conclusion 1, L712) is that 72% (70% as stated in abstract, L22) of total emissions are from facilities that emit less than 100 kg/hr. This is in fact buried in the last table of the supplement, which states the prior emission distribution and shows that 72.7% emissions facilities are from these "small" emitters. The posterior is unchanged from the prior, which is good since MC bootstrapping in this approach shouldn't change the center value.

In this instance, the table that is being referenced (Table S5) represents the posterior (i.e. the estimated individual facility-level emission rates), not the empirical data (i.e., the "prior"). The table's purpose is to easily highlight the information presented in Figure 3 in terms of different emission rate magnitudes and their associated contributions to total oil/gas emissions.

- We edited the caption of Table S5 for better clarity in SI to highlight that it is showing the resulting estimated emissions and does not represent the empirical measurement data, and have also moved the table to the front of the SI Tables.
    - *"Table S1: Breakdown of total oil/gas methane emission for the CONUS in 2021 contributed from different magnitudes of methane emission rates with the corresponding percentage of total facilities responsible for those emissions. These results show a breakdown of the emission distributions curves presented in Figure 3 of the main text."*

A MC bootstrapping technique may be more interesting if applied to randomly select which EF studies to include. For example, if 6 of the 11 studies of facility category "Well Sites" listed in Table S1 were randomly selected for each simulation, then we might assess dependence based on regional dependence of studies, sampling/analytical methodology, etc. Indeed, regional differences are maybe observed, e.g. loss rates of 0.90% for Appalachian and Greater Green River regions (Omara et al, 2018) compared to >4.5% for San Joaquin and San Juan regions, but the variance within the regional populations precludes saying these loss rates are different (based on a Tukey test). Could the Tukey test be run on the $log_{10}(loss \%)$, given that these appear to be lognormally distributed in Figure 1?

We agree with the reviewer that this would be an interesting sensitivity analysis to perform, so we conducted two additional tests (shown in the responses below). The tests examine the impacts of 1) Reducing the number of empirical measurement data to be used in the estimates and 2) Eliminating data from a given oil/gas basin/region (well sites only given limited data on regions for midstream assets. In order, the sensitivity tests show 1) reducing the number of empirical measurements only increases uncertainty bounds but does not affect the overall emission distributions or total emissions estimates 2) excluding data from certain regions does not generally impact our results for emission distributions or total emissions, even for the Appalachian where the majority of our empirical measurement data are located, with the analysis performed on all 9 basins varying the emission distributions by +/-3-4% and our total estimates by +/-6-7%, which is well within our stated uncertainty bounds.

We have since removed the Tukey tests due to this new suggested sensitivity analysis related to the impacts of excluding empirical measurement data from given oil/gas basins. We believe this revised approach better characterizes the uncertainties related to the spatial distribution of measurement data we use in our estimates, since it also includes additional factors such as the relative counts of facilities, differences in oil/gas production, and the number of empirical data available from each region.

- The Tukey test figures have been replaced with a new Figure S9 displaying the resulting changes in total methane emission distribution curves (A) and total methane emissions for the CONUS (B):

[Figure]

○ *"Figure S9: A) Sensitivity analysis of the effects of excluding empirical measurements from a single basin showing the impacts on oil/gas methane emission distributions for the CONUS. 25 emission distribution curves are presented for each basin (colored lines) exclusion scenario with comparisons to the entire dataset of empirical data (black lines). B) Sensitivity analysis of the effects of excluding empirical measurements from a single basin showing the impacts on total oil/gas methane emission estimates for the CONUS. Each box and whisker plot contains 25 estimates of total methane emissions colored according to the oil and gas basin from which empirical measurements were excluded. The black boxplot with red outlines shows the baseline scenario, which has no empirical measurement data removed."*

● New text has been added in the Discussion section:
   ○ [page 28] *"In addition, there are variations in the number of production well site empirical measurements among oil/gas basins (Table S3), although a sensitivity analysis shows that excluding data from individual oil/gas basins does not significantly impact our results (Fig. S9)."*

- Below is a figure illustrating the effects of reducing the sample size of empirical measurement data on total methane emission estimates for the CONUS. 25 iterations are performed for each barplot. "Baseline" refers to the full suite of empirical measurements as we use in our manuscript. Note that we see increased variation in results from a 50% reduction of measurement data relative to our baseline, with no significant change in the average total methane emissions

[Figure]

- Below is a figure illustrating the effects of reducing the sample size of empirical measurement data on methane emission distributions for the CONUS. 25 iterations are performed for each emission distributions. "Baseline" refers to the full suite of empirical measurements as we use in our manuscript. Note that we see increased variation in results from a 50% reduction of measurement data relative to our baseline, with no significant change in the overall emission distributions

2) What is the 95% CI for the total national CH4 emissions?
The 95% confidence intervals for national CH4 emissions are 14.6 (12.7 - 16.8) Tg/yr. We have added these uncertainty ranges in the abstract.

3) Data Availability: Data should be made available in a publically accessible, reliable repository and linked, preferably, through a DOI per EGUsphere instructions.
We agree with the reviewer and are making all the data publicly accessible used for the emission distribution curves presented in Figure 3 (~350,000 rows by 500 columns) with coordinate/facility type/basin level data removed due to data sharing restrictions based on our activity data (i.e., Enverus).

These data are now available for download at Zenodo (link: https://doi.org/10.5281/zenodo.13314532), which now referenced in the Data Availability section.

Ideally, I would also prefer to see a table or reference section in the supplementary that has direct links, references, etc to the data from other studies used in this manuscript. This would be the data references in Table S1, plus Lan et al. 2015.

We have added this information to Table S1 (i.e. links in SI Table S1), including Lan et al. (2015) which was left out due to a clerical mistake and Goetz et al. (2015).

*Specific Comments:*

Would be useful to state what the LOD of Bridger GML is here.
We have included references to the Bridger LOD stated by Kunkel et al. 2023 in this section (i.e., 3 kg/hr).

"1,898 facility-level…" I am a bit confused since Table S1 only sums to 1866 observations.
We thank the reviewer for bringing this to our attention, we neglected to include the Lan et al. (2015) and new Goetz et al. (2015) measurements in this total. This has been fixed in Table S1.

"high-emitting intermittent are included" à "high-emitting intermittent sources are included"
Text has been corrected.

There appears to be a linearly decreasing relationship between the loss % and production rates for well-sites (facility category 5-9). Is this real? Is there a reason to include this in the facility-level model?
Yes, the measurements being shown in this figure are the empirical measurement data we use in our work. This decreasing relationship between production and production-normalized loss rates exists and has been shown/used in prior studies (e.g., Omara et al. 2018). However, the relationship between production and loss rates is weak (i.e., visible in a log/log plot), but useful for better constraining the extrapolation of emission rates to the full population of well sites in CONUS. A more detailed explanation of this relationship is explained in Cardoso-Saldana et al. (2020) (link: https://pubs.acs.org/doi/pdf/10.1021/acs.est.0c03049), but to briefly summarize: emissions from high-producing wells are a combination of production-independent leaks (i.e., fugitive emissions from leaks from pipes, flanges, etc) and production-dependent emissions (i.e., condensate flashing). As the production of a well drops exponentially over time, the associated emissions from production-dependent leaks also drop, whereas the production-independent emissions persist. We utilize this empirically observed relationship between facility level methane loss rate and production to constrain emission estimates for specific production cohorts, where, in general, loss rates are lower for higher producing facilities, and vice versa. This is an important component of our model as the distribution of well site productivity varies across basins.

"… gas flared for 2021 by Elvidge et al. (2016)… efficiencies from Plant et al. (2022)" Are these the correct references? It seems unlikely that Elvidge et al (2016) published gas flaring for 2021.
We thank the reviewer for pointing out this mistake. We have re-written the section to clarify that the Elvidge et al. (2016) reference is meant to provide background on the VIIRS detection instrument and is not being used to draw in actual gas flared values.

"…production well sites that we use in this work generally do not show significant…" à "… basin-to-basin, production well sites in …"
Corrections made

"… Ravikumar et al. (2019) From …" à "…Ravikumar et al. (2019). From…"
Corrections made

What do the error bars represent? 95% CI?
Yes, these bars all represent the associated 95% confidence intervals. We have added new clarifying text in the Figure 4 description.

"our results show the essentiality of expanding beyond solely on super-emitter mitigation". Some sort of grammatical correction needed.
Agreed, corrections made.

It would be nice to provide the sample size of these studies.
Agreed, we have included sample sizes for these studies in other countries.

Appears to be missing a reference to Lan et el. 2015.
Lan et al. 2015 reference added, and number of measured facilities corrected throughout manuscript.

There are several other references used by Omara 2018 not included in this study. (Goetz et al 2015, ERG 2018).

With regards to Goetz et al. 2015, we have since included these data into our dataset of empirical measurement data. The addition of these data (n=3) do not change our results and main findings in any significant way.

We only know of the ERG 2011 study, but we would be happy to investigate a more recent component-level study if another exists. For the ERG 2011 dataset, we decided to exclude it given that it is an older dataset (10+ years) and a compilation of component-level measurements, which we acknowledge in our work may underestimate total facility-level emissions given that there is no guarantee that all emitting components were measured. While we do include some component-level aggregation studies in our work, both of those studies (Riddick et al. 2019, Deighton et al. 2020) provide measurements within the past 10 years. However, we did perform a sensitivity analysis on the effects of including versus excluding the ERG 2011 data and found no change in our model results for both total emissions and the emissions distributions (see below).

- Comparison of total CONUS oil/gas methane emissions when including/excluding ERG 2011 empirical data for 25 estimates each.

[Figure]

- Comparison of emission distribution curves for total CONUS oil/gas methane emissions when including (red)/excluding(black) ERG 2011 values for 25 emission distribution curves each.

[Figure]

The total number of well sites for the Barnett basin is 32 wells less than the sum of the bins. I assume this is the 32 wells measured by Lan et al. (2015) that was not included in Table S1.

That is correct, this was a clerical error on our part. The mistake has been corrected to include the 32 wells from Lan et al. (2015), in addition to the new 3 measurements from Goetz et al. (2015) as previously suggested by the reviewer.

---

## Author Comment (AC3)

**Author response to reviewer comments**

**Anonymous Reviewer #2**

Review of Williams et al., Small emission sources disproportionately account for a large majority of total methane emissions from the US oil and gas sector

The paper examines the role of super emitters in the contribution of different midstream and upstream site categories in the oil and gas sector with regard to CH4 emissions. The authors use published data to create emissions distributions for the different categories as a function of daily gas production rate, using an algorithm that is very similar (identical?) to Omara et al. (2018).

The novel aspect in the present manuscript comes from the framing: Previous papers have stressed that super emitters dominate emissions in a given distribution of emitters, where the super emitters where generally the relatively highest emitters in a given distribution. The new manuscript emphasises the large role of emitters below a fixed absolute emission rate of <100 kg/h, so with this absolute definition, the super emitter category contributes less to total emissions that smaller emitters.

We thank the reviewer for the valuable comments and edits, and we hope the following responses address their concerns.

All of the page references in the responses below reference the attached manuscript with tracked changes included. Text in *"bold blue italics"* references prior text from the manuscript, and text in *"bold red italics"* references new added text.

My biggest concern is that this (and the title of the manuscript) imply a discrepancy between previous and new analyses, which is in fact not the case, and this is not well communicated. I strongly encourage the authors to clarify the change in perspective/framing and where it originates from. It is very clear from Fig 6 of the manuscript that in the category Well sites (<15 boed) there are almost no super emitters according to the absolute definition (>100 kg/h), so it is no surprise that they don't contribute much to emissions. And since this category contributes most to the annual emissions, (Fig 6b) the category strongly lowers the weight on super emitters to the national totals. This should be explained more clearly!

We note the reviewer's concerns on the perspective/framing of the paper and we agree that this is an important point to highlight and explain. Overall, the narrative surrounding oil/gas methane emissions has largely focused on the relationship between facility counts and total emissions, in the sense that a small percentage of the highest emitting facilities (i.e., super-emitters) contribute a disproportionate fraction of total emissions (which we also observe in this work). While these are important findings, our focus is on the relationship of emission rates to cumulative emissions, which we believe to be more critical information relating to methane measurement methodologies, especially with the rise of aerial remote sensing platforms with higher limits of detection. Furthermore, we present these results in the context of

total national US oil/gas methane emissions, accounting for important information such as facility counts and facility types. As the reviewer correctly states, low-producing well sites are the major source of emission in our work, and individually do not emit at high rates, but cumulatively contribute a majority of total oil/gas methane emissions. Low-producing well sites also exhibit the same relationship of facility counts to cumulative emissions, since a small number of the highest emitting low-producing wells contribute the majority of their cumulative emissions. However, these "super-emitting" low producing well sites emit at roughly ~10 kg/hr, which is well below the detection thresholds of satellite point-source imagers, and also some aerial remote sensing platforms.

To better clarify the change in perspective/framing, and to better highlight the contribution of low-producing well sites, we have made the following changes to the main text.

- Text modified in the Abstract [page 1]
  - [page 1] *"We estimate that production well sites were responsible for 70% of regional oil/gas methane emissions, from which we find the well sites that accounted for only 10% of national oil and gas production in 2021, disproportionately accounted for 77% (72-81%) of the total well site emissions."*

- Text modified in the Introduction
  - [page 2] *"Several studies have recognized the importance of a small percentage of high-emitting sites (i.e. "super-emitters") and reported them as accounting for a large fraction of total methane emissions (Brandt et al., 2016; Cusworth et al., 2022; Duren et al., 2019; Sherwin et al., 2024). The emission rate thresholds that characterize these super-emitting facilities are critical information for methane measurement platforms, especially with the rise of remote sensing technologies that face limitations in detecting low-emitting facilities. Aerial and satellite remote sensing technologies have enabled more frequent monitoring of emissions from oil and gas sites and rapid mapping of large areas, although they do face limitations in detection sensitivity. "*

- Text added in Results
  - [page 20] *"Production well sites constitute the bulk of total methane emissions among the facility categories we considered, with most of these emissions contributed from low production well sites. Overall, we find that 77% (72-81%) of well site emissions originated from only 10% of national oil and gas production in 2021 (Fig. S7), highlighting a disproportionately large fraction of emissions relative to production. In terms of individual well site level production values, the same 77% (72-81%) of total cumulative methane emissions were contributed from well sites producing 0.43 kt/yr (0.43-0.45 kt/yr) or lower. For well sites producing 15 boe/day (i.e., 0.13 kt/yr) or lower, which is the production threshold used to define a well site as being marginally producing in previous work (Deighton et al., 2020; Omara et al., 2022), we find that these low producing well sites accounted for 65% (58-69%) of total well site emissions, or 6.4 Tg/yr (4.7-6.8 Tg/yr)."*

- Text added in the Discussion
  - [page 25] *"While detecting and mitigating emissions from super emitters are important (Cusworth et al., 2022; Duren et al., 2019; Sherwin et al., 2024), our results underscore the need to account for oil/gas methane sources emitting at lower rates, as the cumulative*

*contribution of lower-emitting sites accounts for a large majority of emissions across US oil/gas basins. "*

- Text modified in the Conclusions
  - o [page 31] *"3. Production well sites were found to be responsible for 70% of regional oil/gas methane emissions, from which the sites that accounted for only 10% of national oil and gas production in 2021, disproportionately accounted for 77% (72-81%) of the total well site emissions."*

- New Figure S7 relating cumulative well site level production to cumulative well site emissions

[Figure]

- o *"Figure S7: Results from 500 model simulations showing the cumulative methane emissions distribution curves for total well site oil/gas methane emission rates versus the percentage of cumulative combined oil and gas production. Results are ranked first by individual well-site emission rates, and then by well-site combined oil and gas production. The inset table shows the specific percentages of total emissions contributed from production well sites for cumulative well site production values of 1%, 5%, 10%, and 20%. The red arrows correspond to the percentage of total well site emissions contributed from well sites cumulatively producing 10% of total CONUS oil and gas production in 2021."*

My second recommendation is to clearly explain the underlying concept of the model. I read Omara (2018) again, and if I understand correctly, the approach is the following:

1) you use as input i) daily average gas production data for "all" sites and ii) a correlation between production data and measured emission intensity (percentage of production) from a small sub-set of sites, divided into categories.

2) you use a Monte Carlo technique to assign to each site in i) an emission intensity from ii) based on the emission intensity distributions in each category, to calculate hypothetic emissions per site, and then sum sites up in the different categories. You do that many times randomly to get statistically robust data.

In the present manuscript the essence of the model concept is buried in a lot of information on input data. Please state it more clearly.

We acknowledge the lack of clarity and conciseness in the underlying concept of the model highlighted by the reviewer. Briefly, we use separate methods for 1) non-low production well sites, 2) low production well sites and midstream facilities, and 3) VIIRS flare detections. For non-low production well sites, we estimate gas production normalized loss rates for different production bins, which are then converted to methane emission rates using the annual-averaged individual well site gas production values. For midstream facilities and low-production well sites, we estimate the actual emission rates. For VIIRS flare detections, we use the published destruction removal efficiencies from Plant et al. 2022 and the gas flare volumes from VIIRS flares to calculate methane emission rates. In addition to changes to the Figure 2 of the main manuscript, we have also made the following changes in the text

- We have added new text in the first paragraph of the methods to provide a broader description of the essence of the model, and clarify that our methodology is based on multiple studies, and not just Omara et al. 2018.
  - [page 6-7] *"We calculate annual methane emissions from all facility categories (i.e., six production bins of production well sites, T&S compressor stations, G&B compressor stations, and processing plants, and VIIRS flare detections) using a multi-step probabilistic modeling approach adapted from multiple studies (Omara et al., 2018, 2022; Plant et al., 2022) (Fig. 2). Briefly, for each individual facility and VIIRS flare detection in the CONUS for 2021, we estimate an annually averaged methane emission rate using empirical measurement data, and consequently the cumulative distribution of methane emission rates from the aggregation of these individual emission rates. Each emission rate estimate is indexed according to the corresponding replicate (n=500), and we use these repetitions to determine uncertainty for the cumulative methane emission distribution curves. The detailed steps of this process for all facility categories and VIIRS flare detections are described below."*

- New text added in the Methods concerning VIIRS flare detections

  - [pages 8-9] *"For all VIIRS flares detections, we use the total reported volumes of gas flared for 2021 from flares detected using the VIIRS instrument (Elvidge et al. 2016) multiplied by the observed flare destruction efficiencies and percentage of unlit flares from Plant et al. (2022) to*

*calculate annual methane emission rates from this source. As previously stated, our empirical measurements are largely located outside of oil/gas basins where the majority of VIIRS flare detections are located (i.e. Permian, Eagle Ford, and Bakken), but we cannot discount the possibility that there are instances of double-counting flares measured via our ground-based empirical data and those detected by VIIRS. For each VIIRS flare detection, we randomly determine whether it is an unlit or lit flare based on the basin-specific percentages of unlit flares reported by Plant et al. (2022). If a flare is determined to be lit, we use the corresponding basin-specific observed destruction removal efficiencies as reported by Plant et al. (2022) multiplied by the corresponding annual total volume of gas flared and convert to an emission rate. The basin-specific observed destruction removal efficiencies are estimated through a fitted normal distribution using the mean and standard deviations modeled from the 95% confidence intervals presented in Plant et al. (2022). If a flare is determined to be unlit, we use a destruction removal efficiency of 0%. For VIIRS flare detections located outside of the Bakken, Eagle Ford, and Permian basins, we used the total CONUS averaged flaring efficiencies destruction removal efficiencies of 95.2% (95% confidence interval: 94.3 – 95.9%) and percentage of unlit flares of 4.1% as reported by Plant et al. (2022)."*

● Revised Figure 2 to simplify the decision chain of the calculations while also including the methodology for the VIIRS flare detections

[Figure]

● *"Figure 2: Flowchart describing the facility-level estimates, with steps colored according to the specific process and data being used. We note that methane emission rates for flares are*

*calculated using a separate approach from that of production well sites and midstream facilities. Processing plants and T&S compressors are excluded from the determination of whether a facility is a top 5% emitter due to a lack of available empirical measurement data."*

Also, point 2 is making the method rather complicated, and I wonder whether one would not reach the same conclusions by deriving emission factors for the categories based on previous work, and using these emission factors per category for up-scaling.

The primary goal of our work is to characterize emission distributions across the full spectrum of methane emission rates, which requires an estimate of individual methane emission rates that are representative of what is encountered in the field. While we do provide estimates of total emissions at national/basin/aerial scales, in this case, a simple emission factor approach with upscaling by multiplying by number of facilities may produce a reasonable approximation of total methane emissions, but the individual emission rates by facility would not be representative of the actual methane emission rates.

We are attaching a response to anonymous reviewer #1 below, as we believe our response addresses these points:

- Multiplying an average methane emission factor by the number of facilities can produce a rough estimate of total methane emissions, but is not a suitable approach for characterizing facility-level methane emission distributions, which must account for the stochasticity in facility-level methane emissions profiles and related uncertainties (for references that discuss this stochasticity: https://www.nature.com/articles/ncomms14012, https://pubs.acs.org/doi/full/10.1021/acs.est.6b04303, https://pubs.acs.org/doi/full/10.1021/acs.est.8b03535) Developing robust methods for characterizing such distributions at the basin- and national-scale is the focus of this work. While we do present estimates of total emissions estimates at the national/basin/aerial spatial scale, these are a by-product of our methodology and not the main findings, which are the detailed distributions of individual facility-level emission rate and the large majority contribution of total emissions linked to an aggregate of smaller emitting sources (i.e., the distributions presented in Figures 3, 5, and 6).

  If, for example, an EPA GHGI emission factor (e.g., average methane emission rate per facility) and the associated confidence bounds (e.g., standard deviation of the mean) are applied to each individual facility to provide an independent emission rate, and this is repeated for all facilities in the CONUS, this simplified approach would not produce an accurate distribution of emission rates because a representative methane emission factor would still need to account for (i) facilities that may be non-emitting at any one time, (ii) the fact that different facility categories (including different production ranges of well sites) can emit at different rates at any one time, and (iii) the representativeness of facility-level empirical data (and inherent uncertainties in emissions quantification) when compared with the national population of facilities.

For these reasons, we believe a probabilistic modeling approach that accounts for these factors (and others) is essential to assessing emissions distributions and underpins the novel findings we present in this work. Moreover, the conclusions in terms of the specific emission rate thresholds and the aggregate emissions below those and their relative fractions to the total emissions across the US oil and gas upstream and midstream sectors as well as over each individual oil/gas basin has not been produced before based on empirically derived measurement-based analysis, which this study presents as a major step forward in our understanding the dynamics of oil/gas emissions and their source contributions which have important policy implications for measurement and mitigation, as we have discussed throughout the manuscript.

I strongly recommend using SI units, according to Copernicus guidelines (https://www.atmospheric-chemistry-and-physics.net/submission.html#math). I get confused by units like Mcfd, Mcf, boe, and cf3, in the text and in Eq. 1. I realize that these units are used in the O&G industry, but they should not be used in scientific publications. When common SI units are used, the rather trivial unit conversion factors can be omitted in Eq. 1, which would then read:

Emission rate = Gas production * methane content * loss rate * methane density

We agree with the proposed changes. However, we do think that it will be useful to retain the common oil/gas industry terms within the main paper for interested parties in the oil/gas methane science field who are accustomed to industry standards. We have included SI units and relevant conversions in both figure captions (when relevant) and in the main text, and also adjusted equation (1) as suggested by the reviewer. We hope these changes address the reviewers concerns.

- Changes made to Methods
    - [page 7] "*For the highest five gas production bins of producing well sites ranging from 29 to >3,908 Mcf/day (or 0.2 to >27 kt/yr of methane production per year, Figure 1), we use gross gas production normalized loss rates to model the distributions used to calculate methane emission rates from Eq. (1), where the: Loss rate is the fraction of emitted gas relative to gas production, the emission rate is rate of methane emitted from a facility in kilograms per hour, σCH4 is the methane content of the emitted gas which we assume to be 80%, and the gas production is the mass equivalent of natural gas produced in kilograms per hour at 1 atmosphere and 15.6 oC (1 Mcf = 1,000 cubic feet of natural gas = 19.2 kg of methane at 15.6 oC and 1 atmosphere; 1 boe = 1 barrel of oil equivalent = 6 Mcf). For the lowest well site gas production bin of 0 to 29 Mcf/day (i.e, 0 to 0.2 kt/yr of natural gas) and midstream facilities, we use the empirical absolute methane emission rate data as is. This approach is partly based on the methods used by Omara et al. (2022) for the non-low production well site category, which exploits a weak relationship between gross gas production data (which is most accessible in empirical measurement studies) and emission rates to better extrapolate emissions to the entire population of production well sites in the CONUS.*"

        *Loss rate=(Emission rate [kg/hr])/($\sigma_{CH_4}$ × Gas production [kg/hr] )    (1)*

Except for these general points, I find the manuscript well written, but I have a few suggestions to the figures, partly linked to my general recommendations:

I find Fig. 2 complicated and it could be simplified in 2 aspects:

1) remove the "loop over i" going back to the start, and simply state that you do this for all facilities, (then also remove the index i).

2) show separate paths for categories 2, 1&3, and 4-9. I understand that this can be incorporated in an algorithm but this "high level" and rather trivial criterion makes the flow confusing.

These suggestions are very helpful, and we have since incorporated some of these changes into Figure 2 as suggested by the reviewer, including our revised approach for unlit and lit flares. The revised figure 2 was previously shown in a prior response to the reviewer.

In figures 3,5 and 6 the cumulative percentage of emissions is plotted versus emission rate. In many previous studies the cumulative emissions were plotted versus fraction of total sites (ordered from high to low or low to high). These plots usually show the effect of the skewed distributions, namely that the highest emitters in a given distribution contribute most to the emissions. As I mentioned above, this apparent "discrepancy" should be explained, and it may help to also show the cumulative distributions versus fraction of total sites, at least for the category Well sites (<15 boed).

While it is true that many prior studies have shown relationships between the number of sites (ranked by emission rate) versus the cumulative percentage of total emissions, and that the top x% of highest emitting sites contribute a significant percentage of total emissions, our main goal is to define the emission rate threshold at which we define these "high-emitting" sites. For example, we can observe in a newly added Figure S13 that the same relationship between facility counts and cumulative emissions as highlighted in previous studies exists (i.e., small percentage of facilities contribute most to emissions). While this holds valuable information, it does not offer any information regarding the technologies needed to detect these "super-emitting" sites. We can determine that the top 1% of ranked emitting facilities contribute 37% of total emissions, and simultaneously that the bottom 99% of emitting facilities contribute 63% of total emissions. However, these analyses do not offer any information regarding the emission rate of these facilities, which is critical information when developing MRV policies. For example, the top 1% of emitting facilities (contributing ~40% of total emissions) have an average emission rate of 60 kg/hr. The fact that the top x% of facilities contribute a majority of emissions is a valuable insight. However, this offers no information regarding the emission rate detection thresholds necessary to measure/detect these facilities, which is the primary information we are communicating in this work.

In addition to the newly added Figure S13 in the SI, we have also made additional changes in the main text to better highlight this relationship (i.e., cumulative methane emissions versus facility counts or emission rates). We would also point to prior changes made regarding low-production well sites that we believe address the reviewers concerns.

- The following changes were made in the Discussion.
  - [page 25-26] *"Most of our analysis centers around quantifying the percentage contributions of oil/gas methane sources emitting below one discrete emission rate threshold (i.e., <100 kg/hr, per EPA's definition of a super-emitter). We estimate*

*that over 99% of the total oil/gas facilities that we analyze in this work are emitting <100 kg/hr (Fig. S13), which in turn contribute 70% (61 – 81%) of total methane emissions (Fig. 3). The emission rate threshold of 100 kg/hr is relevant to US policy decisions (EPA's Final Rule for Oil and Natural Gas Operations Will Sharply Reduce Methane and Other Harmful Pollution., 2024), but we also illustrate a complete characterization of emissions, which gains importance as newer methane monitoring technologies have different LODs. For example, the effective LOD at high probabilities of detection for available point source imaging satellites of ~200 kg/hr (Jacob et al., 2022) would only be able to quantify 20% (10-32%) of all oil/gas point sources in the CONUS, if the full oil/gas sector was mapped in its entirety, based on our facility-level results. When considering the relationship of facility-level emission rates to total cumulative methane emissions, we find that oil/gas methane emissions in the CONUS are dominated by many low-emitting facilities, which relates directly to methane measurement technologies."*

[Figure]

○ *"Figure S11: Results from 500 estimated facility-level emission distributions showing the cumulative percentages of total methane emissions contributed from facilities emitting below methane emission rate thresholds and colored according to the percentage of total emitting sites ranked by emission rate."*

Technical points:

L127: …. intermittent sources…..

Changes made

L 185 and Eq. 1: It is not appropriate to use the chemical formula CH4¬ as the "methane composition", in eq. 1. Refer to it as" methane content" of the gas and use a proper symbol

Equation has been corrected and simplified

L188: Omara (2020) is not in the reference list, should this be Omara (2018), otherwise add reference
Reference has been corrected

L534: reformulate

- Section has been rewritten, changes noted in a prior response

---

## Author Response (AR2)

**Response to Referee #1**

**Referee-** The authors have re-focused the motivation away from national and regional emission rates and loss rates. Instead, they have expanded on a previous conclusion regarding the use of emission rate distributions to inform us on the ability of different technologies to locate emitting facilities.

This work is still focused on compiling literature emission rates (previously published in Omara 2022, 2023, 2024) which are categorized by facility type and production rate. These emission rate distributions are scaled nationally and by region using infrastructure inventories (Enverus, OGIM database) and validated against remote sensing surveys.

The manuscript is well written and the subject is of suitable content for EGUsphere. The subject is timely as there is active discussion regarding how mitigation funding can most effectively be used to reduce fugitive emissions from O&G. The figures are well designed and informative and have been improved from the first version.

While I do find the focus compelling, I still think the results fairly incremental. The majority of the data and analysis has been published previously (Omara 2022, 2023, 2024). The description of the probabilistic model is still confusing (general comment 1) and results are sometimes presented inappropriately, e.g. the bootstrapping approach provides confidence intervals about the probability of a facility emitting <LOD and not the probability itself.

**Response-**We thank the referee for their comments. We understand that the probabilistic model could still be explained more clearly and have made additional changes to the text in the methods, which include some additions to Figure 1 that also address the bootstrapping procedure also mentioned above. As responded in our previous round of response, we have clarified and addressed the referee's comments that this work is a significant novel work compared to previous Omara et al papers.

Additional changes-
L221: "…we first use bootstrapping with replacement (n=1,000) of our empirical measurement data to simulate the frequency of finding an individual facility emitting above the method LOD…"
L223: "The results of the bootstrapping procedure represent a normal probability distribution from which we estimate the frequency of finding an emitting facility (above the method LOD) with associated uncertainty bounds."
L233: "Similar to the process of determining the frequency of finding an emitting facility, we use the results of the bootstrapping to develop a normal probability distribution that classifies an emitting facility as either a top 5% or bottom 95% emitter."
L242: "Loss rates are used to calculate emission rates for the top five highest production bins of well sites, whereas we directly estimate methane emission rates for the well sites in the lowest production cohort (Figure 1), and for midstream facilities excluding VIIRS flare detections."

Revised Figure 1 with the estimated mean frequency of finding a facility emitting <LOD listed.

[Figure]

Figure 1 caption: "(...)The estimated mean frequency of finding a facility emitting below the method LOD is shown in inset red text at the bottom of each boxplot. We show absolute emission rates (kg/h) rather than normalized loss rates (%) for the lowest cohort of production well sites (...)"

**Response to Referee #3**

**Referee-** I require major revisions to the manuscript before I could recommend this manuscript for publication:
I recommend narrowing the scope of the study to focus on depth instead of breadth.
Don't try to bootstrap to basins, let alone the entire nation. This is unfounded and damages the credibility of the manuscript. Also, the most important findings in the paper can be supported without bootstrapping.
Instead, just analyze the aggregate distributions of the small sample size. To my knowledge, this is the first time the ground studies have been aggregated, and there are very valuable learnings from that alone.

**Response-** We agree with the referee that a more in-depth analysis of the empirical measurement data would be valuable and have added further details in Figure 1 that displays the estimated frequency of finding a facility emitting below the method LOD. We have further included direct analyses of the empirical measurement data in this work, as well as the sensitivity tests on the impacts of varying method detection limits and sample sizes as shown in Figures S8 and S9, and the general outlines and summary statistics of the empirical data in Tables S2 and S3.
We believe that a comprehensive assessment of facility-level emissions distributions must account for the diversity in facility-level characteristics that are inherent in each measurement data (e.g., for well sites: data collected over multiple years include a wide range in production rates and different facility-level emission profiles for various facility sub-types) and can vary from basin to basin. By incorporating oil and gas activity data at the basin and national level, our modeling approach, that takes into account of the compiled direct measurement data, makes it possible to effectively integrate these facility-level emission data over the population of methane emitting facilities, and draw more meaningful robust conclusions regarding the mean national- and basin-level distributions and potential implications for methane mitigation. Similar approaches have been used in the past (e.g., Alvarez et al,, 2018; Omara et al. 2018, 2022, 2024; Zavala-Araiza et al. 2015), in recognition of the limitation of simply consolidating empirical data from multiple studies without accounting for underlying factors that could impact each facility's emission rates, including the likelihood of a facility emitting below the detection limit or emitting at the "super-emitter" level.

**Referee-** Figure 1 and the related analysis are highly valuable if they can be trusted. I would like to see the authors provide more in-depth and transparent description of the data curation process. I can't tell by the descriptions of what was included and omitted whether there was cherry picking to support a foregone conclusion. I agree the with the authors' prioritization of studies that included zero emissions, but are there other ground-based studies that were considered and not included? Were all of the data from those studies used without omission? If all this checks out, then Figure 1 is very valuable. The only thing more I'd want to see from this data is what happens to the average loss rates in each category when the zero-emission measurements are included (which they should to give a true loss rate picture of the category).

**Response-** We agree that the data curation process could be explained more clearly and have added new text in the Methods that notes only one study that was not considered due to a combination age and being a component-level study (i.e., chance of "missing" emission sources). All other relevant studies were included to the best of our knowledge.

L123: "Only one study was excluded from our analysis (ERG 2011) due to a combination of age and a focus on component-level measurements."
We account for the influence of facilities with zero emissions in this analysis through the determination of the likely frequency of finding a facility emitting below the method LOD (i.e., the bootstrapping approach). If a facility is classified as emitting below the method LOD through our probabilistic modelling approach, it is then assigned an emission rate through resampling of all empirical measurement data below the method LOD (including zero measurements). Therefore, the loss rates from these BDL measurements are included in our analysis, meaning that the average loss rates would not change. For our results to be robust, it is important that we provide the best possible representation of what magnitude of emission rates would be encountered at individual oil/gas facilities, which requires that all relevant empirical measurement data are considered in our analysis.

**Referee-** Straight up analysis of the ground measurement cumulative emission distribution without any bootstrapping. Show the data so we can see how small the sample looks, especially if you want to break it down by basin and facility category. From that alone, what does that aggregate data tell us about emissions below <100 kg/hr. I can only assume the authors' thesis would remain unchanged: most of the emissions measured by the ground techniques come from emitters <100 kg/hr. If you want to use random resampling of the distributions to generate uncertainty, then that could work. The reach to say that finding is representative of a whole basin or nation is far too big.

**Response-** All the non-zero empirical emission rate data is shown in Figure 1 of the main manuscript, with the zero emission rates accounted for in the total number of emission rate data shown above each section of the box plots. We would also highlight that the dataset we use (~1,900 measurements) is the largest aggregated dataset of ground-based measurements from oil/gas facilities in the CONUS to date (to the best of our knowledge).

For information to referee, we include a figure below that shows the empirical distribution of the four main types of oil/gas facilities, which show agreement with our estimated distributions we present in the paper. For example, the percentage of cumulative emissions from well sites emitting <100 kg/h is 79% in our empirical measurement data, versus 90% (80-100%) in our estimated distributions. However, we note that the approach used in the paper is designed to account for the differences in individual facility characteristics using data such as the individual well site production data, for example, where site population and production rate characteristics vary within and among basins.

As we noted above, the resulting emissions distribution from a simple consolidation of all the empirical data, which are collected from multiple basins with varying distribution of facility types and emission profiles, does not comprehensively represent the population mean emission distributions, which is a more robust metric for understanding implications for methane mitigation. A "straight-up" analysis of the empirical emissions distributions overlooks these important nuances in the data, the effects of which are further likely compounded by our limited sample size, which we acknowledge in the manuscript. Our models and analyses account for these factors, and they provide a comprehensive picture of the mean facility-level emissions distributions.

[Figure]

**Referee-** Rigorously and comprehensively compare the cumulative distribution from the small-ground-measurement sample size to those of the remote sensing distributions. The emissions scientific community needs such cross comparisons very badly, even with limited sample size and undetermined representativeness (this is all we have). I would like to see a detailed and transparent description of comparisons with Bridger, Carbon Mapper, and Methane

Air. I don't see how it could be possible that the ground distribution matches all three. That's okay, it will be one more piece to fill out the fuller picture.

**Response-** We agree that there is a need to perform these types of intercomparisons of aerial/satellite measurements to ground-based approaches – which is exactly what we have included in this paper, especially in the context of cumulative methane emission rate distributions, as this is the focus in the existing Section 3.4 and existing Figure 7 (see below) of our work. We would also highlight Figure S3 (see below) that shows a more complete intercomparison of the continuous emission rate distributions between this work and Bridger data from two studies, which is possible given the much lower 90% POD of Bridger compared to CarbonMapper or MethaneAIR. For our comparisons, we do indeed find good agreement on the distributions of emissions for all three aerial measurement platforms across multiple oil/gas basins, due in part to the steps we take to ensure a robust comparison of our results to those from aerial campaigns (i.e., restricting analysis to the same source types, ensuring the same spatial domains are used, focusing on emission rate thresholds that correspond to the measurement platform). We see the agreement between our results and the aerial campaigns as both validation of our probabilistic modelling approach using ground-based measurements, and that low-emitting methane sources in aggregate contribute a significant fraction of total oil/gas methane emissions. We have added additional text in the Methods that better describes this process, per referee's comment.

L273: "To perform these comparisons, we restrict our estimates and the results from other aerial/satellite studies to spatial domains of interest (e.g., an oil/gas basin boundary or the overflown domain from an aerial sampling campaign), and to specifically compare estimates of oil/gas methane emissions from the facility categories we are investigating in this work."

[Figure]

Existing Figure 7 – Shows strong agreement for all measurement platforms across different oil/basins, except for the Appalachian basin where the subtraction of non-oil/gas sources and pipeline emissions produces some additional uncertainties.

[Figure]

Existing Figure S3 – Comparison of continuous emission distribution curves to Bridger sampling campaigns which shows good agreement in emissions distributions above 3 kg/hr.